# The Crohn's disease polymorphism, *ATG16L1* T300A, alters the gut microbiota and enhances the local Th1/Th17 response

Sydney Lavoie[1,2], Kara L Conway[3], Kara G Lassen[4,5], Humberto B Jijon[3], Hui Pan[6], Eunyoung Chun[1,2], Monia Michaud[1,2], Jessica K Lang[1,2], Carey Ann Gallini Comeau[1,2], Jonathan M Dreyfuss[6], Jonathan N Glickman[7,8], Hera Vlamakis[4], Ashwin Ananthakrishnan[3], Aleksander Kostic[6,9†*], Wendy S Garrett[1,2,4,10†*], Ramnik J Xavier[3,4†*]

[1]Department of Immunology and Infectious Diseases, Harvard T. H. Chan School of Public Health, Boston, United States; [2]Department of Genetics and Complex Diseases, Harvard T. H. Chan School of Public Health, Boston, United States; [3]Gastrointestinal Unit, Center for the Study of Inflammatory Bowel Disease, Massachusetts General Hospital, Boston, United States; [4]Broad Institute of Harvard and MIT, Cambridge, United States; [5]Center for Computational and Integrative Biology, Massachusetts General Hospital, Boston, United States; [6]Joslin Diabetes Center, Boston, United States; [7]Department of Pathology, Harvard Medical School, Boston, United States; [8]Beth Israel Deaconess Medical Center, Boston, United States; [9]Department of Microbiology and Immunobiology, Harvard Medical School, Boston, United States; [10]Department and Division of Medical Oncology, Dana-Farber Cancer Institute and Harvard Medical School, Boston, United States

**\*For correspondence:**
Aleksandar.Kostic@joslin.harvard.edu (AK);
wgarrett@hsph.harvard.edu (WSG);
xavier@molbio.mgh.harvard.edu (RJX)

†These authors contributed equally to this work

**Abstract** Inflammatory bowel disease (IBD) is driven by dysfunction between host genetics, the microbiota, and immune system. Knowledge gaps remain regarding how IBD genetic risk loci drive gut microbiota changes. The Crohn's disease risk allele *ATG16L1* T300A results in abnormal Paneth cells due to decreased selective autophagy, increased cytokine release, and decreased intracellular bacterial clearance. To unravel the effects of *ATG16L1* T300A on the microbiota and immune system, we employed a gnotobiotic model using human fecal transfers into *ATG16L1* T300A knock-in mice. We observed increases in *Bacteroides ovatus* and Th1 and Th17 cells in *ATG16L1* T300A mice. Association of altered Schaedler flora mice with *B. ovatus* specifically increased Th17 cells selectively in *ATG16L1* T300A knock-in mice. Changes occur before disease onset, suggesting that *ATG16L1* T300A contributes to dysbiosis and immune infiltration prior to disease symptoms. Our work provides insight for future studies on IBD subtypes, IBD patient treatment and diagnostics.
DOI: https://doi.org/10.7554/eLife.39982.001

## Introduction

Crohn's disease (CD) and ulcerative colitis (UC), the two main forms of inflammatory bowel disease (IBD), are characterized by chronic relapsing inflammation of the gastrointestinal tract (*Podolsky, 2002*; *Turpin et al., 2018*). The etiology of IBD is complex, as host genetics, the gut microbiota and environmental exposures all contribute to disease pathogenesis (*Xavier and Podolsky, 2007*; *Garrett et al., 2010a*). A breakdown in the ability of a genetically susceptible host to respond

**eLife digest** Trillions of bacteria live inside the human gut. From helping to digest our food to producing vitamins, these bacteria can have a big impact on our health, yet some people tolerate these bacteria better than others. In some cases, the body reacts badly to its own bacteria, stimulating an over-exuberant immune response. The gut becomes too inflamed, causing pain and diarrhoea, which could lead to an inflammatory bowel disease such as Crohn's disease or ulcerative colitis. The symptoms of inflammatory bowel disease can vary from person to person, and how someone responds to treatment can be as individual as the symptoms as well.

The causes of inflammatory bowel disease are complex; our genes, immune system and gut bacteria all play a role. Previous research has found hundreds of mutations in our genes that increase a person's risk of developing inflammatory bowel disease. No single mutation is the root cause for every one person with inflammatory bowel disease, and individuals with these mutations may not even develop the condition. For those who do develop the disease, certain immune cells can be found in high numbers, such as the white blood cell known as Th17 cells. People with inflammatory bowel disease may also house different bacteria compared to someone with a healthy gut. In some cases of inflammatory bowel disease, there are elevated amounts of one type of bacteria called *Bacteriodes*. It is not clear how these mutated genes, the types of bacteria that live inside our gut, and the immune response are all related.

Lavoie et al. focused on a mutated gene, known as *ATG16L1T300A*, which increases risk of Crohn's disease in humans, and the experiments compared mice that had this human mutated gene with those that did not. The mice started off germ-free, meaning that they did not have any gut bacteria. Lavoie et al. then exposed the mice to samples of human stools, which contain gut bacteria, and after a month they analysed the guts of the mice. On average, the mutant mice had more *Bacteriodes* and Th17 cells in their guts than the normal mice. However, none of the mice developed inflammatory bowel disease, suggesting that changes to gut bacteria and immune cells may occur before the disease can be diagnosed.

Together these findings show how just one mutated gene affects the bacteria and immune cells in the gut; but there are hundreds of other known mutations linked with inflammatory bowel disease. By unravelling the effects of more of these mutations, scientists could begin to learn more about the causes of this condition, and potentially improve its treatment options.
DOI: https://doi.org/10.7554/eLife.39982.002

appropriately to the gut microbiota may lead to an overactive local immune response (*Sartor, 2008*; *Eckburg and Relman, 2007*) initiating the chronic cycle of intestinal inflammation core to IBD. Many genes within IBD loci are directly involved in pathways controlling the sensing and innate responses to bacteria (*Xavier and Podolsky, 2007*; *Jostins et al., 2012*). The relatively longstanding observation that there is an absence of intestinal inflammation in several gnotobiotic mouse models of spontaneous colitis maintained under germ-free housing conditions supports this idea (*Elson et al., 2005*; *Sellon et al., 1998*). Furthermore, data from IBD patients demonstrating that diversion of the fecal stream greatly improves symptoms (*Rutgeerts et al., 1991*; *McIlrath, 1971*) as well as reduces inflammatory cytokine levels (*Daferera et al., 2015*) also lends plausibility to this concept.

Dysbiosis of the gut microbiota, including alterations in frequency, diversity and richness of microbial populations (*Manichanh et al., 2006*; *Ott et al., 2004*), has been associated with IBD (*Morgan et al., 2012*; *Frank et al., 2007*; *Willing et al., 2009*). For example, a reduction in the abundance of the phylum Firmicutes, including the genus *Clostridium* (*Rajilić-Stojanović et al., 2013*) as well as Proteobacteria and Actinobacteria, has been associated with IBD (*Frank et al., 2007*). In contrast, there is variation in the abundance of *Bacteroides* in IBD (*Swidsinski et al., 2002*; *Swidsinski et al., 2005*; *Gophna et al., 2006*; *Rehman et al., 2010*; *Chu et al., 2016*), including an increase in *Bacteroides fragilis* (*Swidsinski et al., 2005*; *Gophna et al., 2006*). Additionally, certain microbes, including *Ruminococcus gnavus*, are able to thrive in the high levels of oxidative stress in the inflamed gut and are associated with increased disease activity (*Hall et al., 2017a*). A population-based metagenomics analysis found associations between fecal levels of secreted proteins and microbiome composition (*Zhernakova et al., 2016*). Additionally, the unique ileal transcriptome of

CD patients is associated with alterations in Firmicutes (*Haberman et al., 2014*), implicating a role for genetics in shaping the microbiota. Genome wide association studies have shown that host genetic variants play a role in microbial dysbiosis (*Hall et al., 2017b*). Further supporting the idea that these microbial alterations could be the driving force behind the immune response, microbial changes have been observed prior to onset of disease symptoms and specific microbiota signatures are associated with disease location (*Imhann et al., 2018*). While data suggest a dynamic interplay between the microbiota, the immune system and disease pathophysiology (*Palm et al., 2015*), how dysbiosis manifests and more specifically how individual species are affected by host genetics is less well understood. Because there is substantial diversity in disease location, symptoms and genetic susceptibility in IBD, understanding how specific microbial alterations occur will be important for identifying disease subtypes as well as developing precision medicine treatment options. Genetic diversity among patients is a compounding factor with respect to disease phenotype, suggesting that specific risk loci may play a role in underlying microbial community makeup.

Genome-wide association studies have linked over 230 genetic risk loci to increased IBD risk (*Jostins et al., 2012*; *Liu et al., 2015*; *de Lange et al., 2017*; *Luo et al., 2017*). The microbiota and disease severity have also been linked to genetic risk (*Moustafa et al., 2018*). Genetics can affect the composition of the human gut microbiome (*Hall et al., 2017b*). Additionally, IBD is associated with a variety of risk alleles that affect immune activation in response to microbial recognition and handling (*Davies and Abreu, 2015*). Many studies have linked IBD to genetic risk variants in *NOD2* (*Turpin et al., 2018*), which is involved in recognition of bacterial muramyl dipeptide, but one study has linked the presence of multiple NOD2 variants to increased levels of *Enterobacteriaceae* (*Knights et al., 2014*). A single nucleotide polymorphism (SNP) in *ATG16L1* (re2241880, Thr300Ala) is associated with an increased risk of CD (*Hampe et al., 2007*; *Rioux et al., 2007*). ATG16L1 functions in autophagy, a cellular recycling system that aids in the sequestration of intracellular bacteria (*Cooney et al., 2010*; *Travassos et al., 2010*). *ATG16L1* hypomorphic mice have defects in anti-microbial peptide secretion by Paneth cells (*Cadwell et al., 2008*) and *ATG16L1*-deficient macrophages show increased secretion of pro-inflammatory cytokines (*Saitoh et al., 2008*). Conditional deletion of *ATG16L1* specifically in epithelial cells reduces autophagy and renders mice more susceptible to *Salmonella enterica* serovar Typhimurium infection and systemic translocation (*Conway et al., 2013*). The Thr300Ala (T300A) variant displays enhanced degradation via active caspase 3 resulting in a reduction in the levels of ATG16L1 protein and reduced levels of autophagic flux (*Murthy et al., 2014*; *Lassen et al., 2014*). Moreover, microbial signatures are associated with variants in the gene for NOD2 (*Xavier and Podolsky, 2007*), which interacts with ATG16L1 (*Sartor, 2008*). T300A knock-in mice display defects in Paneth cells and goblet cells (*Lassen et al., 2014*). More recently, T300A has been shown to directly affect lysozyme secretion from Paneth cells in mice in the context of infection (*Bel et al., 2017*). These data suggest that T300A may play a role in shaping the response to gut microbes. Specifically, this may occur by loss of autophagy due to enhanced degradation of ATG16L1 T300A and altered anti-microbial peptide secretion. These effects in turn may affect microbial composition. While many of these studies have observed differences in mice in the context of infection, the effect of T300A on microbial composition has not been addressed. Furthermore, while these data suggest that alterations in the makeup of gut microbial communities may arise from the presence of certain risk alleles and may shape the downstream immune response; this has yet to be formally demonstrated.

A dysfunctional response to the microbiota can lead to adaptive immune cell infiltration in the gut during colitis (*Feng et al., 2010*; *Lodes et al., 2004*) and is a key contributor to disease pathology. Initially, studies suggested a role for Th1 cells in CD (*Parronchi et al., 1997*) whereas Th2 cells were viewed as responsible for inflammation in UC (*Inoue et al., 1999*). However, a more nuanced understanding of the role of T cells in IBD has lead to the implication of Th17 cells in CD pathogenesis as well. Th17 cells are characterized by the expression of the transcription factor RORγt[+] and produce high levels of the cytokines IL-17 and IL-22. High levels of these cytokines have been found in the gut mucosa of CD patients as compared to healthy controls (*Fujino et al., 2003*; *Andoh et al., 2005*). More recently, IFN-γ, a Th1 cell cytokine, IL-17 and IL-22 have been shown to be increased in the affected ileum compared to the uninfected ileum of CD patients (*Li et al., 2017*). Because it is still unclear whether dysbiosis precedes inflammation, the initiation of an adaptive response to the gut microbiota in IBD still needs to be fully elucidated. Herein, we have utilized 16S rRNA gene amplicon sequencing, shotgun metagenomic sequencing, and analysis of immune cell populations in

the gut to demonstrate that the risk allele T300A affects gut microbial composition in mice, both during inflammation under conventional housing conditions, and also in gnotobiotically-housed mice that harbor microbial communities derived from stool specimens from IBD patients, or in mice associated with *Bacteroides ovatus* with a limited, defined microbial community. Furthermore, we have shown that these differences can affect the immune response in the gut lamina propria (LP) prior to the manifestation of disease.

## Results

### T300A conventionally-housed specific-pathogen-free (SPF) mice display altered gut microbial composition at steady state

Previous studies have shown that IBD risk alleles can affect microbial composition (*Knights et al., 2014*). Because ATG16L1 is involved in the handling of intracellular bacteria, we asked whether the IBD risk allele T300A could affect the gut microbiota in conventionally-housed SPF wild type (WT) vs. T300A knockin mice (*Lassen et al., 2014*) at steady state. We conducted 16S rRNA gene amplicon surveys on stool samples from WT and T300A mice (*Figure 1*). We observed overall alterations in the microbiota of T300A mice compared to WT mice (*Figure 1a*). WT and T300A mice showed distinct separation by t-distributed stochastic neighbor embedding (t-SNE) analysis (*Figure 1b*) and we saw an increase in the order Bacteroidales (*Figure 1c*). Overall, these data suggest that T300A mice have alterations in the gut microbiota at steady state and that the abundance of the order Bacteroidales may be influenced by the presence of the T300A IBD risk allele in mice.

### T300A conventionally-housed mice display altered gut microbial composition and an increased abundance of bacteroidetes including *Bacteroides ovatus* during gut inflammation

Given that the alterations observed in the human gut IBD microbiota are often studied in the context of gut inflammation and injury, we sought to determine whether the T300A allele would influence the gut microbiota in mice during a state of large intestinal injury and inflammation modeled using a chemical-based perturbation, as T300A mice do not spontaneously develop intestinal inflammation. We treated WT and T300A mice with 2.5% dextran sulfate sodium (DSS) in the drinking water for 7 days followed by 7 days of regular drinking water. We conducted 16S rRNA gene amplicon surveys on stool specimens after this two-week intervention from WT and T300A mice (*Figure 2*). We observed a decrease in the phylum Firmicutes and an increase in the phyla Bacteroidetes, Proteobacteria, and Cyanobacteria in T300A mice as compared to WT mice (*Figure 2a*). The microbiota of WT mice separated from cage-matched T300A mice by t-distributed stochastic neighbor embedding (t-SNE) analysis (*Figure 2b*). These data support that T300A plays a role in shaping the gut microbiota in the context of intestinal injury and recovery. When we looked specifically at the phyla Bacteroidetes, there was an increased abundance in T300A mice compared to WT mice (*Figure 2c*) suggesting that the T300A risk allele may specifically enhance *Bacteroides spp*. In particular, *B. ovatus* was increased significantly in T300A mice (*Figure 2d*). We analyzed the data using PICRUSt in order to infer metabolic function and found increased glycosaminoglycan degradation in samples from T300A mice (*Figure 2—figure supplement 1*), a process which is uniquely attributed to Bacteroidetes (*Koropatkin et al., 2012*). The degradation of glycosaminoglycans by Bacteroidetes has the potential to affect gut barrier function by reducing the thickness and integrity of the mucus layer and in turn contributes to bacterial translocation and host immune system activation. This observation has implications for the mechanism by which increased abundance of Bacteroidetes can contribute to disease pathogenesis in IBD. Additionally, we also observed increased weight loss in T300A mice from day 8 through 13 following treatment with DSS compared to WT mice, suggesting that these alterations in the microbiota are associated with heightened disease susceptibility (*Figure 2e*).

### Human stool donor disease status shapes gut microbiota composition in mice

Next, we sought to determine whether the differences observed in the microbial community in T300A conventionally-housed mice during inflammation were specific to the mouse microbiome or whether the host T300A genotype could also influence the composition of donor human stool. We

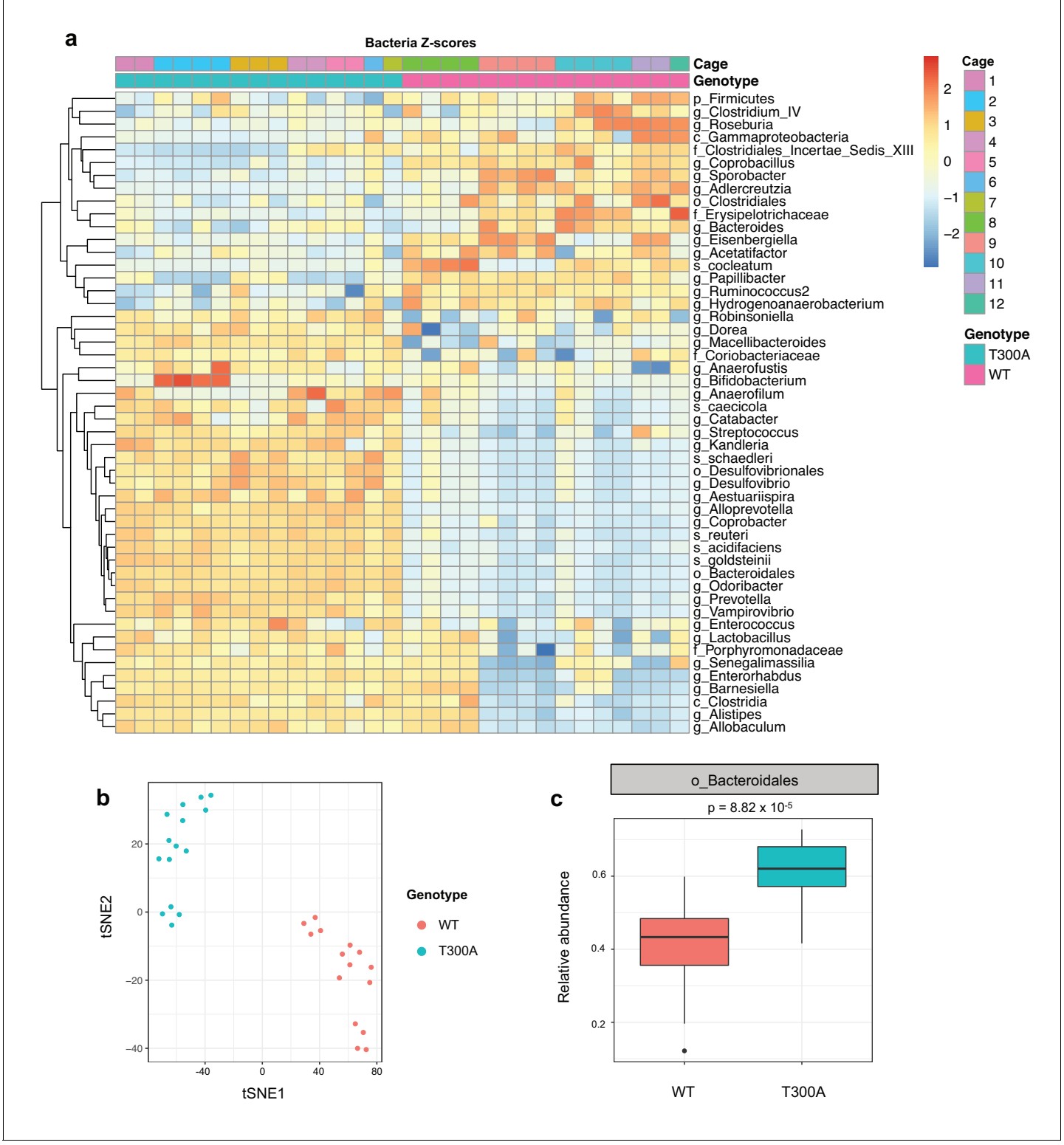

**Figure 1.** Altered microbiota in conventionally-housed WT vs T300A mice. 12 week old WT and T300A mice were conventionally-housed in a specific pathogen free (SPF) facility. Stool samples were collected and analyzed by 16S rRNA gene amplicon sequencing. (a) Population analysis of 16S rRNA gene amplicon sequencing from stool samples from SPF WT vs. T300A mice. (p = phylum, c = class, o = order, f = family, g = genus, s = species) Bacteria Z-scores generated by limma. (b) t-Distributed Stochastic Neighbor Embedding (tSNE) plot analysis of WT vs. T300A SPF mouse stool samples. (c) Box plot analysis of the order Bacteroidales in WT vs. T300A SPF mouse stool samples.

DOI: https://doi.org/10.7554/eLife.39982.003

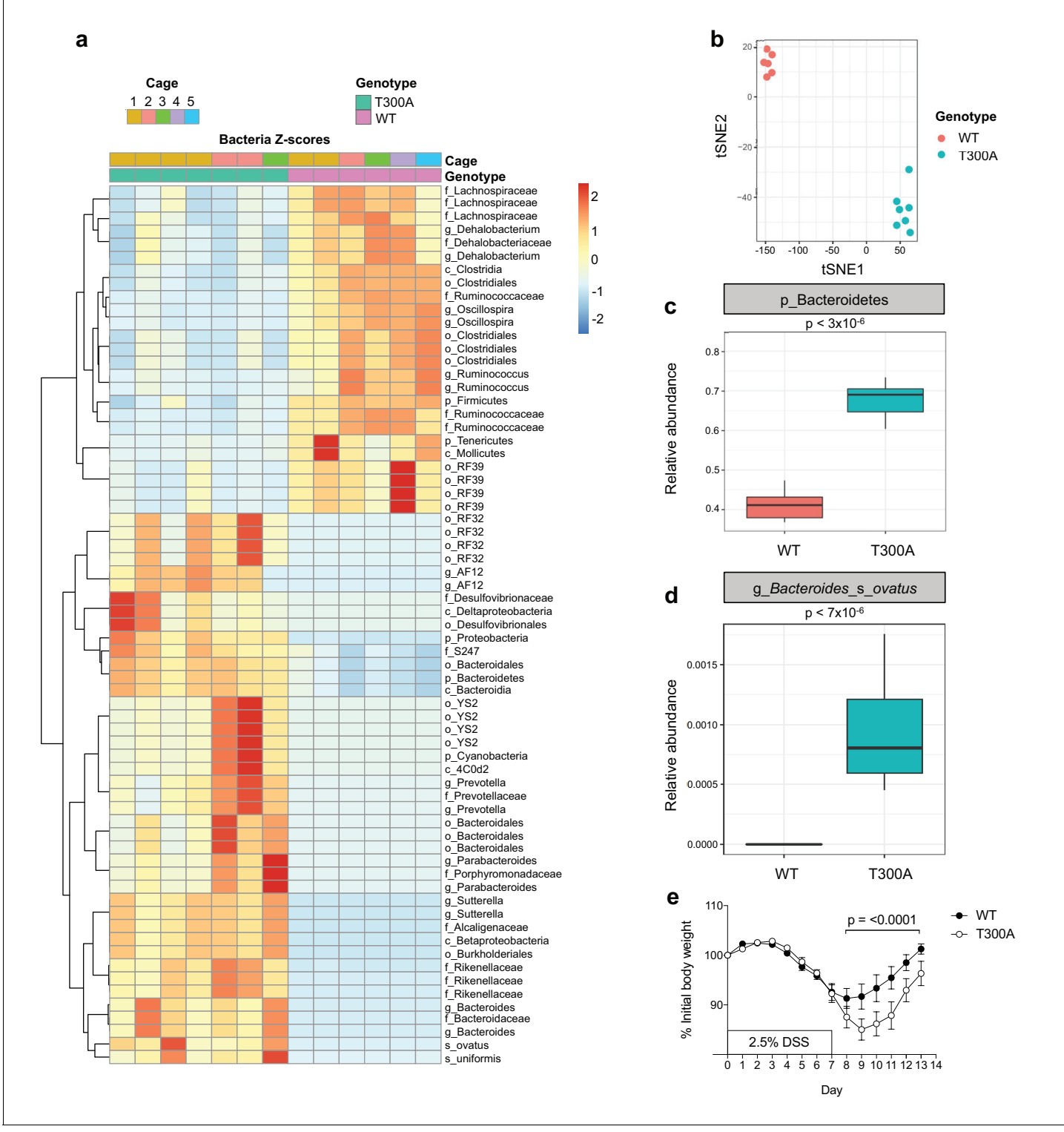

**Figure 2.** Altered microbiota in conventionally-housed WT vs T300A mice with gut inflammation. 12 week old WT and T300A conventionally-housed SPF mice were treated with 2.5% DSS for 7d followed by 7d of regular drinking water. (**a**) Population analysis of 16S rRNA gene amplicon sequencing from stool samples from WT vs. T300A mice. (p = phylum, c = class, o = order, f = family, g = genus, s = species) Bacteria Z-scores generated by limma. (**b**) t-Distributed Stochastic Neighbor Embedding (tSNE) plot analysis of WT vs. T300A stool samples post DSS treatment. (**c**) Box plot analysis of the phylum Bacteroidetes in WT vs. T300A mice post DSS treatment. (**d**) Box plot analysis of *B. ovatus* in WT vs. T300A mice post DSS treatment. (**e**) Percent initial body weight of WT (n = 12) and T300A (n = 15) mice treated with 2.5% DSS for 7d followed by 7d of regular drinking water. Day 8–13 time points, p = < 0.0001. Two-way ANOVA with Tukey's post-hoc test.

*Figure 2 continued on next page*

*Figure 2 continued*

DOI: https://doi.org/10.7554/eLife.39982.004

The following source data and figure supplement are available for figure 2:

**Source data 1.** Source data for *Figure 2*.

DOI: https://doi.org/10.7554/eLife.39982.006

**Figure supplement 1.** T300A affects the microbiome gene function profile.

DOI: https://doi.org/10.7554/eLife.39982.005

obtained a limited number of human stool samples from patients with a genotype of WT or T300A and with active or inactive disease status to associate germ-free (GF) mice. It should be noted that the T300A allele is a relatively common mutation in healthy individuals (*Hampe et al., 2007*), thus our healthy control is heterozygous for T300A based on the availability of stool samples. First, to verify that patient stool samples would maintain a composition based on human donor disease status within the mouse gut, we orally inoculated WT and/or T300A gnotobiotically-housed mice with stool samples from two CD patients in remission (donor genotype T300A or WT, termed 'inactive CD'), a CD patient with active inflammation (genotype T300A, termed 'active CD'), an ulcerative colitis (UC) patient with active inflammation (genotype T300A, termed 'active UC'), or a healthy volunteer (genotype heterozygous T300A (AG), termed 'healthy control') (*Figure 3a*). We compared the phylogenetic compositions from 16S rRNA gene amplicon sequencing data from the original human donor stool samples with phylogenetic composition analyses from our shotgun metagenomic data using MetaPhlAn from stool samples from GF mice associated with the indicated human stool samples. We observed that samples from these mice maintained diversity based on patient disease status by PCA analysis (*Figure 3a*). Individual mice associated with the same patient stool clustered together with the original samples from human donor samples used for the GF associations. This observation suggested that the differences in the human microbiome from patient samples were maintained within the mouse gut.

## T300A alters gut microbiota composition in gnotobiotically-housed mice associated with active CD stool

To determine whether the presence of T300A could alter the gut microbiota, we orally inoculated human stool from individual IBD patients with active disease (one with CD or one with UC) into gnotobiotically-housed WT or T300A mice. Four weeks after oral inoculation, we assessed the stool microbial composition by metagenomic sequencing and MetaPhlAn followed by PCA analysis (*Figure 3b*). We observed a difference in WT vs. T300A mice by PCA in mice that received stool from a CD patient with active disease but not in mice inoculated with stool from a UC patient with active disease (*Figure 3b*). These data suggest that differences in the active CD microbiota in mice are due to the presence of the risk allele T300A as opposed to microbial population drift due to human stool transfer into mice alone. When we examined the microbial composition of WT vs. T300A mice associated with stool from IBD patients with active disease, we observed an increase in the genus *Bacteroides* in T300A mice that received stool from patients with active CD as compared to WT mice (*Figure 4a*). *B. ovatus* was also increased in T300A mice that received stool from patients with active CD (*Figure 4b*), further supporting the idea that T300A has the potential to drive increases in *Bacteroides spp*. We did observe high levels of *Bacteroides* in mice that received stool from patients with active UC but no significant difference between WT and T300A mice (*Figure 4b*). We did not see major significant changes to the microbiota in WT vs. T300A mice associated with active UC stool. These data suggest that the human CD microbiota may be more susceptible to the presence of T300A, driving larger differences in microbial populations compared to mice associated with stool from patients with active UC.

## Presence of T300A in humans increases *Bacteroides*

Previous reports on the abundance of Bacteroidetes show conflicting results in humans (*Hold et al., 2014*), with some studies reporting an increased abundance while others report a decreased abundance in patients with IBD. Therefore, we investigated whether the increase in *Bacteroides* in the microbiota of T300A mice was also seen in IBD patients with the risk allele T300A (*Figure 5*). We

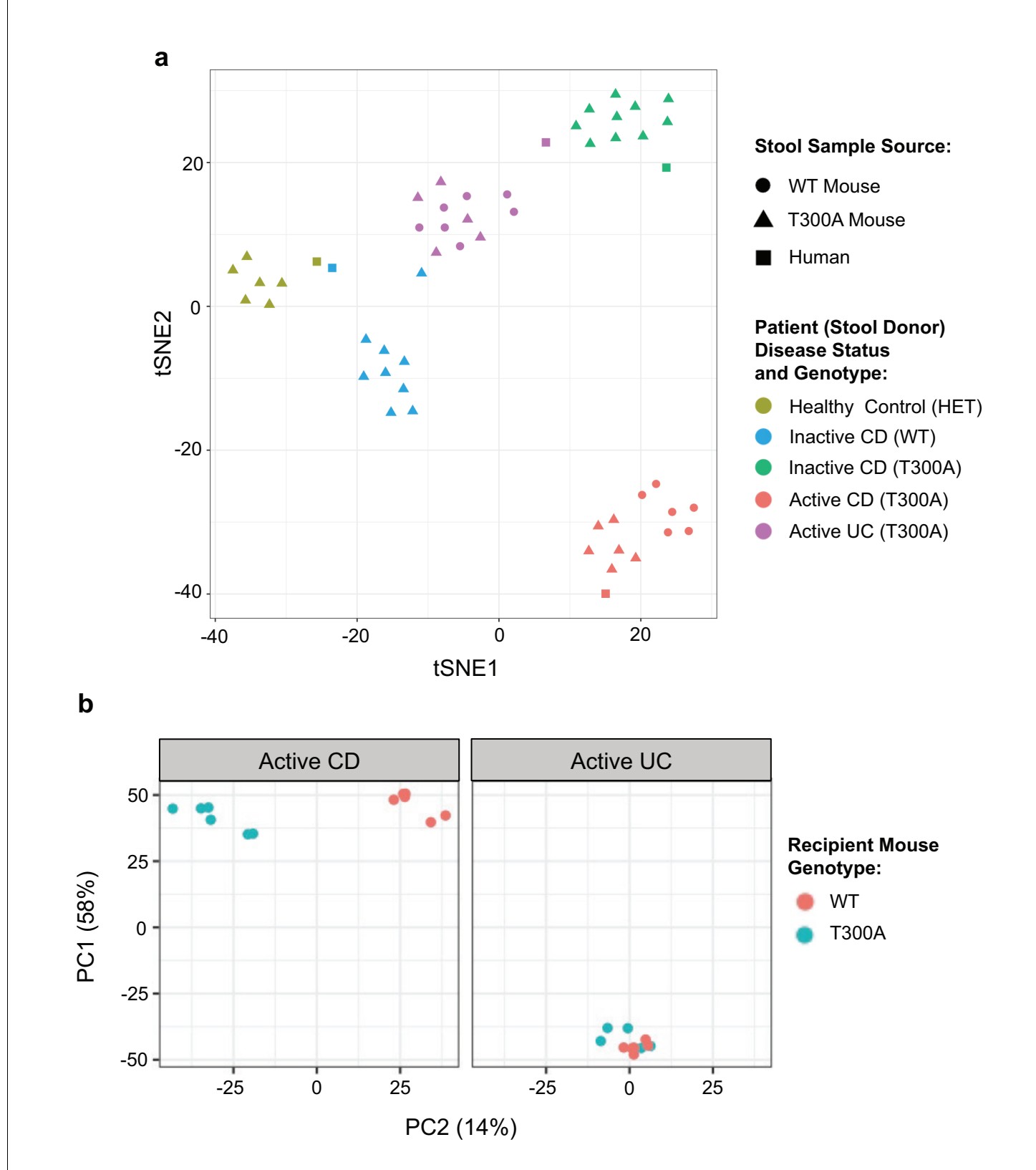

**Figure 3.** Distinct gut microbiota composition in mice associated with human stool. GF mice were associated with human stool samples and analyzed after four wks. (a) tSNE dimensionality reduction analysis based on genus-level abundances of stool samples from human patient stool donor (square) or GF WT (circles) and T300A (triangles) mice four wks post association with 50 mg human stool from patients: healthy control, genotype Het (yellow),

*Figure 3 continued on next page*

*Figure 3 continued*

inactive CD, genotype WT (blue), inactive CD, genotype T300A (green), active CD, genotype T300A (red), or active UC, genotype T300A (purple). Each symbol represents data from an individual mouse. (**b**) PCA plot of stool samples from WT vs. T300A mice four wks post association with either active CD (left) or active UC (right) human stool. Each symbol represents data from an individual mouse.
DOI: https://doi.org/10.7554/eLife.39982.007

utilized patient genotype data and stool microbiome data available from a previous study (*Jostins et al., 2012*). When we compared the microbiota from IBD patients with WT (AA) genotype versus heterozygous patients (AG) versus T300A (GG) genotype, we saw a trend toward an increase in the genus *Bacteroides* (*Figure 5a*). We also observed a trend toward an increase in the *B. fragilis* group (*Figure 5b*) and a significant increase in the species *Bacteroides caccae* (*Figure 5c*). In contrast, we observed a decrease in *Clostridia* and an increase in *Gammaproteobacteria* (*Figure 5—figure supplement 1*) both of which have been observed in numerous human IBD studies (*Frank et al., 2007*; *Gophna et al., 2006*; *Rehman et al., 2010*) and pre-clinical colitis models (*Rooks et al., 2014*; *Garrett et al., 2010b*). These data show that the presence of T300A has the potential to reconfigure the human microbiota and that the presence of a heterozygous genotype displays an intermediate phenotype with *Bacteroides spp*, suggesting that even a mild alteration in ATG16L1 can have an effect on bacterial abundance. This association of the risk allele T300A and alterations in the microbiota have not previously been described. This link between genotype and gut microbial phenotype in humans is important for furthering our understanding of the basis of IBD pathogenesis. Whether or not genotype and the microbiota are also associated with differences in immune populations in humans, prior to IBD disease onset or recrudescence is yet to be determined.

## T300A selectively alters gut immune cell populations in mice associated with active CD patient stool

We sought to discern whether the alterations we observed in the gut microbiota of T300A mice were also associated with changes in immune cell populations in the gut of gnotobiotic WT versus T300A mice associated with patient stool. Crohn's disease has been linked to an increase in both

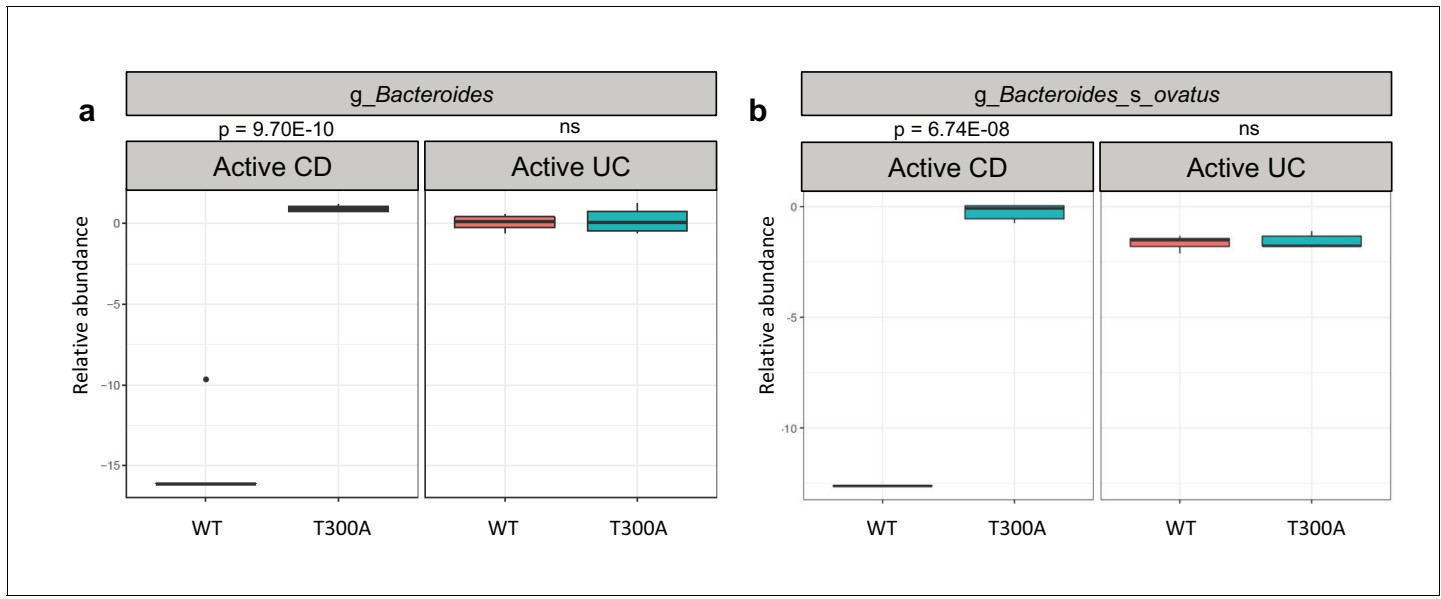

**Figure 4.** T300A alters gut microbiota composition in GF mice associated with human stool. Metagenomic population analysis (MetaPhlAn) on stool samples (four wk time point) from WT vs. T300A GF mice associated with human IBD stool samples (**a**) Box plot of the genus *Bacteroides* from stool samples from mice associated with active CD or active UC stool samples. (**b**) Box plot of *B. ovatus* from stool samples from mice associated with active CD or active UC stool samples. P-values were generated using Benjamini-Hochberg false discovery rate. (**a, b**) Y-axis is log10 relative abundance.
DOI: https://doi.org/10.7554/eLife.39982.008

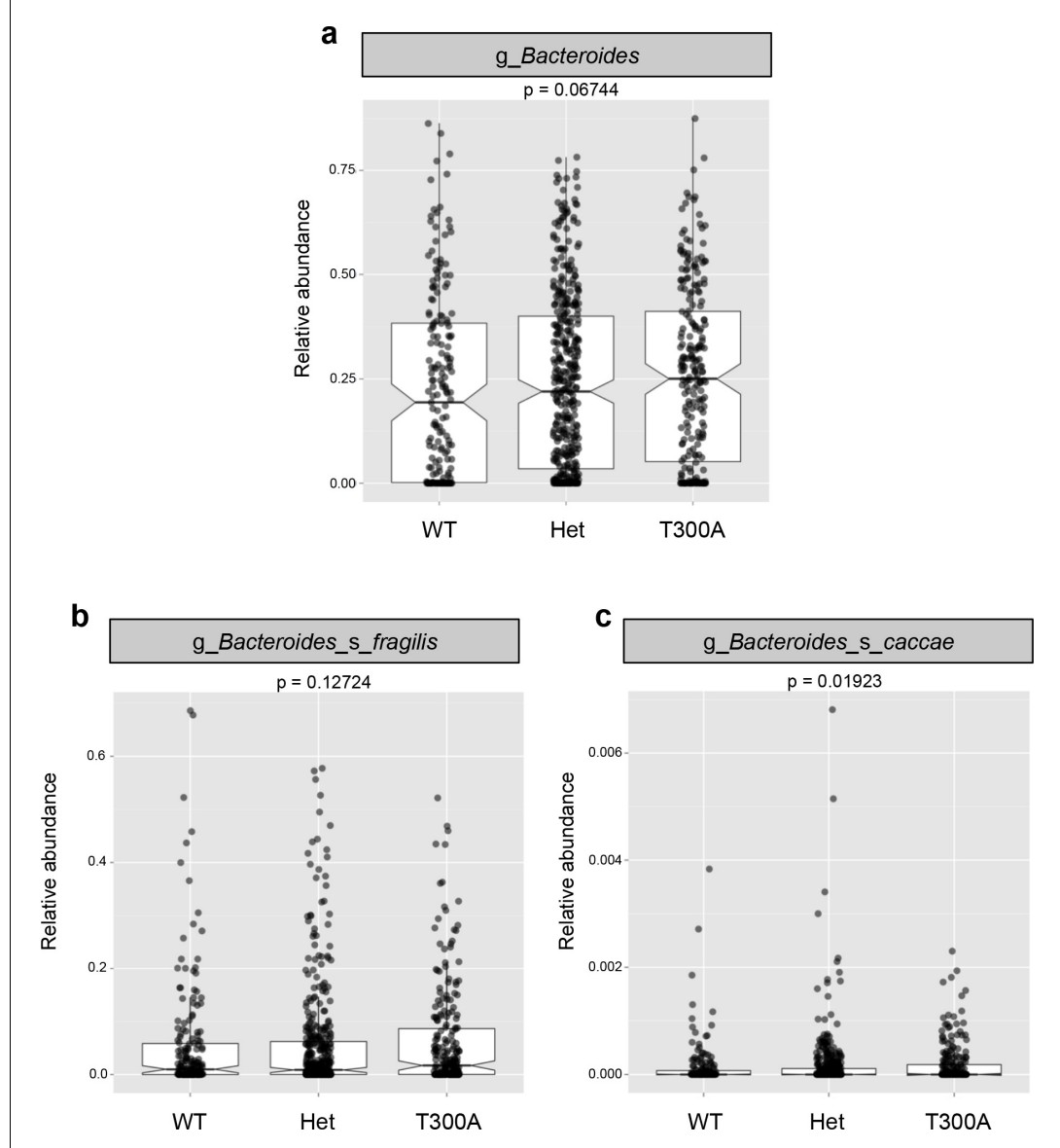

**Figure 5.** The presence of T300A in humans increases *Bacteroides*. 16S rRNA gene amplicon data from human stool samples from Jostins et al. 2012 (*Jostins et al., 2012*) cohort were analyzed for differences in the microbiota associated with the presence (T300A – GG), absence (WT – AA) or heterozygous (Het – AG) genotype. (g = genus, s = species). (**a**) Relative abundance of *Bacteroides*. (**b**) Relative abundance of *B. fragilis*. (**c**) Relative abundance of *B. caccae*.

DOI: https://doi.org/10.7554/eLife.39982.009

The following figure supplement is available for figure 5:

**Figure supplement 1.** The presence of T300A in humans reduces *Clostrida spp.* and increases *Gammaproteobacteria*.

DOI: https://doi.org/10.7554/eLife.39982.010

Th17 cells and Th1 cells in the gut (*Parronchi et al., 1997*; *Fujino et al., 2003*; *Andoh et al., 2005*; *Li et al., 2017*). Th17 and Th1 cells are characterized by the expression of the transcription factors RORγt and T-bet respectively. Therefore, we analyzed RORγt[+] (Th17) and T-bet[+] (Th1) T cells in addition to Foxp3[+] regulatory T cells (Treg), as well as GATA-3[+] (Th2) cells by flow cytometry from mice associated with stool from patients with active CD or active UC (*Figure 6*). T cells were gated as the frequency of CD3[+]CD4[+], live, CD45[+], lymphocytes (*Figure 6—figure supplement 1*). Myeloid cells can also play a role in inflammation during IBD, and loss of ATG16L1 in myeloid and dendritic cells leads to enhanced colitis and increased production of inflammatory cytokines (*Zhang et al.,*

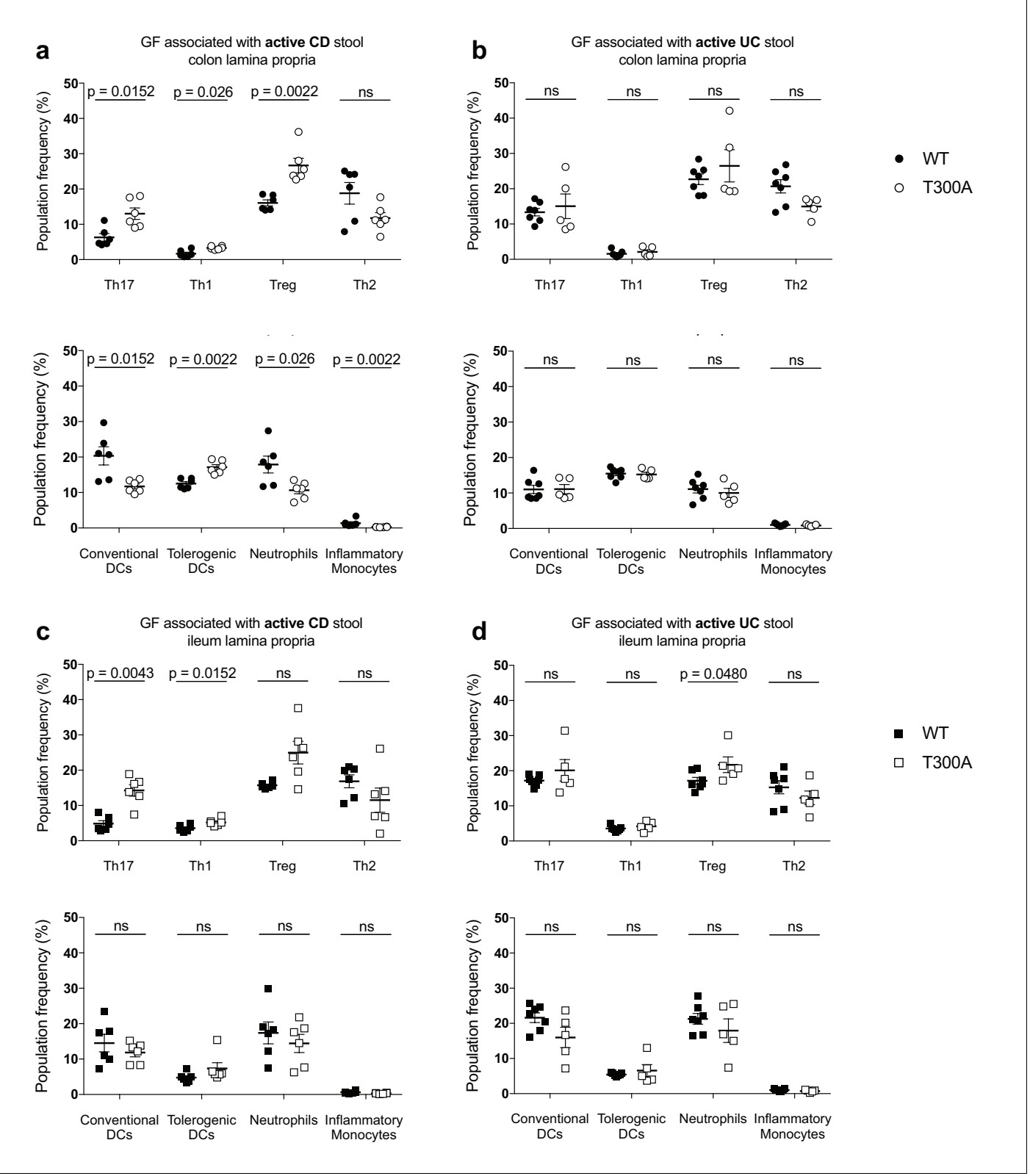

**Figure 6.** Alterations in T cell and myeloid cell populations in the gut of T300A mice associated with stool from patients with active Crohn's disease. LP flow cytometry of T and myeloid cell populations from GF mice associated with 50 mg of frozen stool from a patient with Crohn's disease (CD, WT $n$ = 6 and T300A $n$ = 6) or ulcerative colitis (UC, WT $n$ = 7 and T300A $n$ = 5) for four wks. T cells gated as the frequency of RORγt$^+$ (Th17), T-bet$^+$ (Th1), Foxp3$^+$ (Treg) or GATA-3$^+$ (Th2) cells out of CD3$^+$CD4$^+$ cells and myeloid cells gated as the frequency of CD11c$^+$MHCII$^+$ cells out of CD45$^+$ cells

*Figure 6 continued on next page*

*Figure 6 continued*

(conventional DCs), CD11b⁻CD103⁺ cells out of CD11c⁺MHCII⁺ cells (tolerogenic DCs), CD11b⁺GR-1$^{int}$ cells out of CD45⁺ cells (neutrophils), and CD11b⁺GR-1$^{hi}$ cells out of CD45⁺ cells (inflammatory monocytes). CD3⁺CD4⁺ and myeloid cell populations were gated on single, live, CD45⁺ lymphocytes. (a) GF +CD stool colonic LP T cells (top) and myeloid cells (bottom) (b) GF +UC stool colonic LP T cells (top) and myeloid cells (bottom). (c) GF +CD stool ileal LP T cells (top) and myeloid cells (bottom) (d) GF +UC stool ileal LP T cells (top) and myeloid cells (bottom). Data are plotted as the mean ± SEM. ns = not significant. Mann-Whitney *U* test.

DOI: https://doi.org/10.7554/eLife.39982.011

The following source data and figure supplements are available for figure 6:

**Source data 1.** Source data for *Figure 6*.
DOI: https://doi.org/10.7554/eLife.39982.015
**Figure supplement 1.** Representative T cell flow cytometry gating strategy.
DOI: https://doi.org/10.7554/eLife.39982.012
**Figure supplement 2.** Representative myeloid cell flow cytometry gating strategy.
DOI: https://doi.org/10.7554/eLife.39982.013
**Figure supplement 3.** Microbiota changes and immune infiltration occur in the absence of disease.
DOI: https://doi.org/10.7554/eLife.39982.014

*2017*). Thus, we analyzed the following myeloid cell populations: CD11c⁺MHCII⁺ conventional dendritic cells (DCs), CD11b⁻CD103⁺ tolerogenic DCs, CD11b⁺GR-1$^{int}$ neutrophils, and CD11b⁺GR-1$^{hi}$ inflammatory monocytes by flow cytometry (*Figure 6*). Conventional DCs, neutrophils and inflammatory monocytes were gated out of live, CD45⁺ cells and tolerogenic DCs were gated out of conventional DCs (*Figure 6—figure supplement 2*). We examined the frequency of T cell populations and myeloid cell populations in the lamina propria (LP) of the colon and ileum from WT and T300A mice associated with stool from patients with either active CD or active UC (*Figure 6*). We observed increased frequencies of Th17 and Th1 T cells exclusively in the LP of the colon and ileum from T300A mice that received stool from patients with active CD (*Figure 6a and c*) and not in the LP of the colon or ileum of mice that received stool from patients with active UC (*Figure 6b and d*). These data support an association between alterations in the gut microbiota driven by the risk allele T300A and increased frequencies of Th17 and Th1 T cells in the gut LP. We also found an increase in the frequency of Tregs in the LP of the colon from T300A mice associated with stool from patients with CD (*Figure 6a*) and a minor increase in the ileum in T300A mice associated with stool from patients with UC (*Figure 6d*). The only differences we observed in myeloid cells occurred in the LP of the colon from mice associated with stool from patients with active CD (*Figure 6a*). We saw a reduced frequency of conventional DCs and a slight increase in the frequency of tolerogenic DCs along with a decrease in neutrophils and inflammatory monocytes in T300A mice (*Figure 6a*). Why some of these populations are reduced is yet to be determined. Overall these data suggest that the presence of the T300A allele in mice can enhance Th17 and Th1 cells in the LP of the colon and ileum and alter the frequency of DCs, neutrophils, and inflammatory monocytes in the presence of stool from patients with active CD. It should be noted that histology-based grading of the colon and small intestine from mice associated with human IBD stool showed no signs of colitis, enteritis, or any intestinal histopathology (*Figure 6—figure supplement 3*). This suggests that an alteration in the microbiota composition and alterations in gut immune cells occur before the onset of disease as assessed by histology. Whether specific microbes are responsible for differences in individual subsets of immune cells in the absence of disease is not well understood.

## T300A selectively enhances Th17 cells in the gut of mice with a restricted, defined microbiota in the presence of *Bacteroides ovatus*

Previous studies support that *Bacteroides spp.* can drive a Th17 response in the colon (*Tan et al., 2016*), suggesting that the increase we observe in Th17 cells in the LP of the colon from T300A mice could be driven by the increase in *Bacteroides* in these mice (see *Figure 4*). Therefore, we assessed whether the presence of *B. ovatus,* a species that we observed increased in abundance in T300A mice associated with active CD stool, would selectively alter gut immune populations in T300A mice. To test this hypothesis we associated GF and altered Schaedler flora (ASF), a restricted, defined microbial community of eight bacterial members, WT and T300A mice with ~10⁹ CFU of *B. ovatus* 8483 for 3 weeks. Because it has been shown that housing of mice in separate cages can affect the

composition of the gut microbiota, WT and T300A mice were co-housed during the association of all studies performed with *B. ovatus* 8483. Information on caging, age, and sex of mice for all studies can be found in *Supplementary file 1*. We found no difference in any immune populations in WT or T300A GF mice or GF mice associated solely with *B. ovatus* 8483 (*Figure 7*). However, T300A ASF mice showed an increase only in Th17 cells in the LP of the colon and ileum when *B. ovatus* 8483 was present (*Figure 8b and d*), but there was no difference in ASF mice in the absence of *B. ovatus* 8483 (*Figure 8a and c*). Besides a slight decrease in Th1 cells in the ileum of T300A ASF mice in the presence of *B. ovatus* 8483 (*Figure 8d*), there were no other changes in the immune populations analyzed. These data suggest that the presence of the T300A allele and additional members of the microbiota can increase the presence of Th17 cells in the gut. This increase in frequency of Th17 cells in T300A ASF mice with *B. ovatus* 8483 also occurred in the absence of colitis, enteritis, or any intestinal histopathology (*Figure 8—figure supplement 1*), or differences in the level of colonization of WT and T300A mice with *B. ovatus* 8483 (*Figure 8—figure supplement 2*). This demonstrates that the differences in Th17 cells in ASF T300A *B. ovatus* 8483 mice were not due to an increase in colonization of *B. ovatus* but rather were most likely due to the presence of the T300A allele alone.

## The presence of *Bacteroides ovatus* increases the expression of *Il23p19* in T300A ASF mice

The increase in Th17 cells in the gut of T300A ASF mice associated with *B. ovatus* prompted us to look at expression levels of the IL-23 specific subunit, IL-23p19, in the LP of the colon and ileum from T300A mice (*Figure 9*). IL-23 induces IL-17 producing cells that can induce inflammation (*Langrish et al., 2005*) and is essential for the production of IL-17 in T-cell mediated colitis (*Yen et al., 2006*). Additionally, antibodies targeting IL-23 have been utilized as a treatment for IBD and antibodies specific for IL-23p19 show promising clinical trial results (*Moschen et al., 2018*), suggesting that IL-23 plays an important role in IBD pathogenesis. We found that in T300A ASF mice, the presence of *B. ovatus* 8483 increased expression levels of *Il23p19* in the LP of both the colon (*Figure 9a*) and ileum (*Figure 9b*). Overall, these results demonstrate that the presence of *B. ovatus* 8483 in the context of the T300A allele and a minimal microbiota can enhance both IL-23 expression and Th17 cell infiltration into the gut before the onset of disease symptoms in mice.

## Discussion

The role of the gut microbiota in IBD pathogenesis is well established, but how and when microbial dysbiosis occurs during disease progression is less well understood. The identification of IBD genetic risk loci involved in the processing and handling of microbes provides more definitive clues into disease pathogenesis. However, we still do not fully understand how host genotype or particular SNPs influence microbial community assembly, community composition, or function. The reduced barrier integrity associated with defects in Paneth cell anti-microbial peptide secretion in T300A hypomorphic mice (*Cadwell et al., 2009*) and T300A knock-in mice (*Conway et al., 2013*), and increased production of inflammatory cytokines by ATG16L1 deficient innate immune cells in response to infiltrating bacteria (*Saitoh et al., 2008*) can activate an adaptive immune response to the gut microbiota. However, whether genetic risk loci directly affect specific gut microbial population frequency is not known. Here we show that the presence of the CD risk allele *ATG16L1* T300A in both mice and humans can shape the gut microbiota. More specifically, it can enhance the abundance of *Bacteroides*, including *B. ovatus*, as well as increase the local Th1 and Th17 response in the gut LP in T300A mice associated with stool from a patient with active CD but not stool from a patient with active UC.

We found that minimal microbiota ASF T300A mice associated with *B. ovatus* 8483, specifically increased Th17 cells, suggesting that the other differences we saw in Treg and myeloid cell populations in T300A mice associated with stool from a patient with active CD may be due to the presence of other species in the CD microbiota. While we observe increases in other microbial populations, including Proteobacteria and Cyanobacteria in conventional-housed T300A subjected to chemically-induced inflammatory conditions, the increase in Bacteroidetes was common between both DSS-treated conventionally-housed mice and gnotobiotically-reared human stool associated mice, suggesting that T300A could potentially be a driver of increased Th17 and Th1 T cell populations in the gut LP via increased *Bacteroides spp.* When T300A ASF mice were associated with *B. ovatus* 8483,

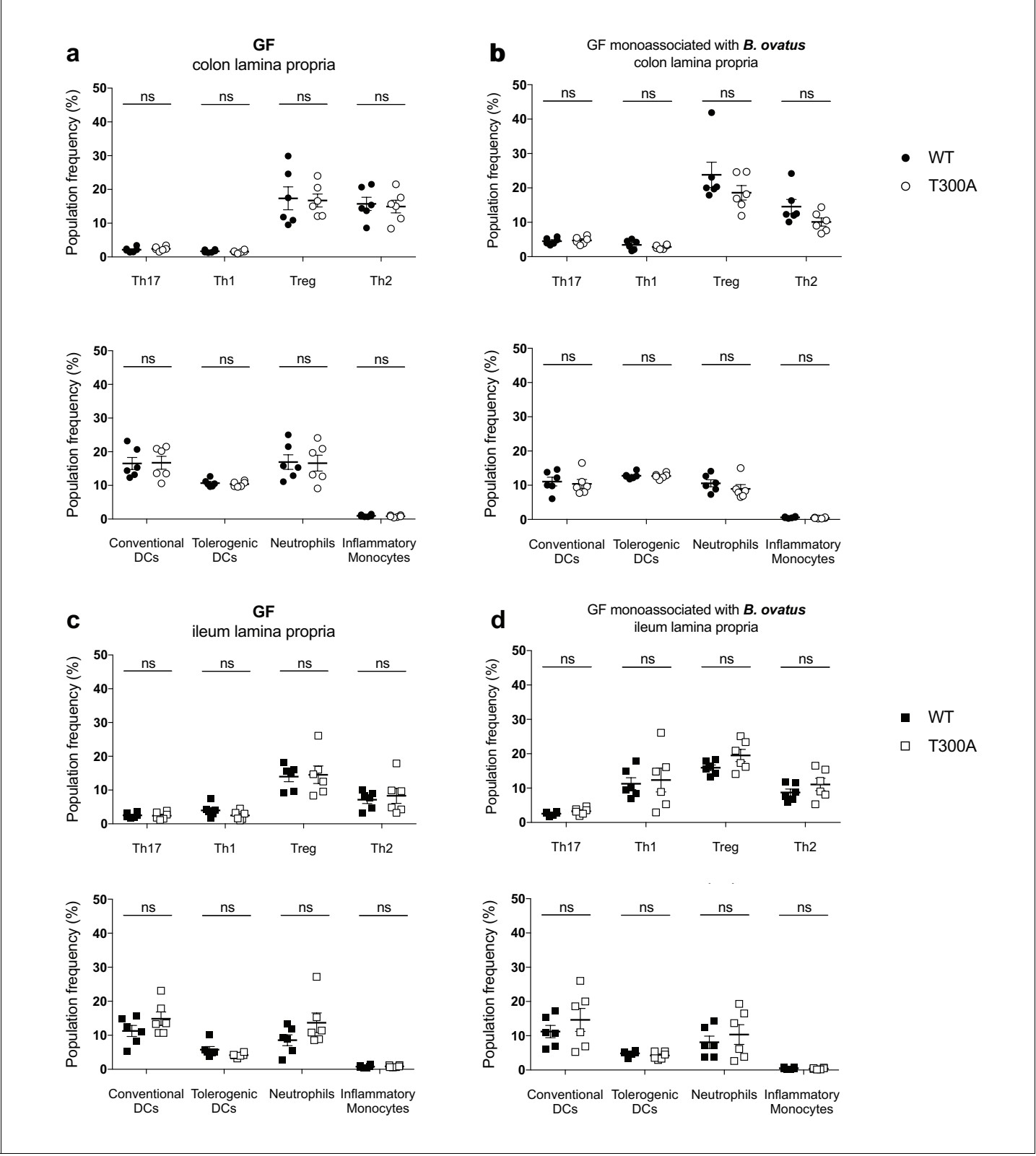

**Figure 7.** No change in immune cell population frequency in the gut of GF or GF mice monoassociated with *B. ovatus* 8483. LP flow cytometry of T and myeloid cell populations from GF mice (WT *n* = 6, T300A *n* = 6) or GF mice associated with ~10⁹ CFU of *B. ovatus* 8483 for three wks (WT *n* = 6, T300A *n* = 6). T cell populations are gated as the frequency of RORγt⁺ (Th17), T-bet⁺ (Th1), Foxp3⁺ (Treg) or GATA-3⁺ (Th2) cells out of CD3⁺CD4⁺ cells and myeloid cell populations are gated as the frequency of CD11c⁺MHCII⁺ cells out of CD45⁺ cells (conventional DCs), CD11b⁻CD103⁺ cells out of

*Figure 7 continued on next page*

Figure 7 continued

CD11c⁺MHCII⁺ cells (tolerogentic DCs), CD11b⁺GR-1$^{int}$ cells out of CD45⁺ cells (neutrophils), and CD11b⁺GR-1$^{hi}$ cells out of CD45⁺ cells (inflammatory monocytes). CD3⁺CD4⁺ and myeloid cell populations were gated on single, live, CD45⁺ lymphocytes. (a) GF colonic LP T cells (top) and myeloid cells (bottom) (b) GF +*B. ovatus* 8483 colonic LP T cells (top) and myeloid cells (bottom) (c) GF ileal LP T cells (top) and myeloid cells (bottom) (d) GF +*B. ovatus* 8483 ileal LP T cells (top) and myeloid cells (bottom). Data are plotted as the mean ± SEM. ns = not significant. Mann-Whitney *U* test.

DOI: https://doi.org/10.7554/eLife.39982.016

The following source data is available for figure 7:

**Source data 1.** Source data for *Figure 7*.
DOI: https://doi.org/10.7554/eLife.39982.017

we did not observe differences in levels of colonization, suggesting that the increase in the frequency of Th17 T cells in the gut LP was independent of the level of colonization. Why we did not find differences in immune populations in GF WT and T300A mice monocolonized with *B. ovatus* 8483 is intriguing as it suggests that this increase in Th17 cells requires the presence of a more complex microbiota. Future work will be needed to elucidate how the immune response to *B. ovatus* changes in the presence of additional microbial species and in the context of distinct host genotypes associated with IBD. Specifically, what is the nature of the co-occurrence relationships between *B. ovatus* and other members of a microbiota necessary for Th17 T cell expansion? Alternatively, are there a series of one-on-one host-microbe relationships in tandem or sequentially that are required for Th17 T cell expansion in the LP of the colon and ileum?

Previous studies have demonstrated the ability of individual *Bacteroides spp.* to induce IBD in genetically susceptible mice (*Bloom et al., 2011*) and others have shown that *Bacteroides spp.* can induce IL-17 response in the gut (*Tan et al., 2016*). Exactly how T300A could enhance *Bacteroides* specifically is yet to be determined. We found that expression of a subunit of IL-23, *Il23p19*, a cytokine known to increase IL-17 production from T cells, was increased in both the LP of the colon and the ileum from T300A mice associated with *B. ovatus* 8483. Other studies have shown that *B. fragilis* enterotoxin can induce formation of autophagosomes (*Ko et al., 2017*). Another report demonstrated that ATG16L1 is required to generate a regulatory T cell response to *B. fragilis* outer membrane vesicles (*Chu et al., 2016*). Because T300A is susceptible to caspase 3 cleavage and is preferentially targeted for degradation compared to WT ATG16L1 (*Turpin et al., 2018*), reduced xenophagy levels could allow for an outgrowth of *Bacteroides*, and an insufficient regulatory T cell response necessary to dampen Th17 population expansion. ATG16L1 and Nod2 (another CD risk allele) have been shown to interact in an autophagy-dependent antibacterial pathway (*Homer et al., 2010*), suggesting that defects in either pathway could affect *Bacteroides* abundance. For example, Nod2 is essential for preventing the expansion of *Bacteroides vulgatus* in mice (*Ramanan et al., 2014*). We also saw that T300A in humans was associated with a trend in increased levels of *Bacteroides* including the *B. fragilis* group and a significant increase in *B. caccae* which is closely related to *B. ovatus* (*Sakamoto and Ohkuma, 2011*). An intermediate phenotype was seen in people with the heterozygous (AG) genotype. More research, especially in a prospective cohort of at-risk individuals, will be needed to determine whether IBD patients with T300A have an increase in *B. ovatus* prior to disease onset and whether this contributes to disease pathogenesis.

IBD symptoms can vary in nature and severity as well as disease location. While some patients will respond to treatment, others will not, suggesting that subtypes of the disease may be dependent on both genetics and the microbiota. Healthy individuals and CD patients may harbor multiple risk alleles, compounding the effect of specific alleles on the microbiota, and further studies will be needed to address the complexity of genetic associations with gut microbial composition. However, understanding how risk alleles can shape both the microbiota and the underlying immune response, in tandem, can provide insight into identifying subsets of the disease as well as inform diagnosis and treatment decisions. We have shown that the risk allele *ATG16L1* T300A, a SNP associated with an increased risk of CD, contributes to dysbiosis in mice, specifically an increase in *Bacteroides* among other alterations, and correlates with an enhanced Th1 and Th17 immune response in the gut LP. These changes precede the onset of disease in human stool microbiome associated mice, suggesting that microbiota changes induce inflammatory cell population shifts in the gut. These results shed

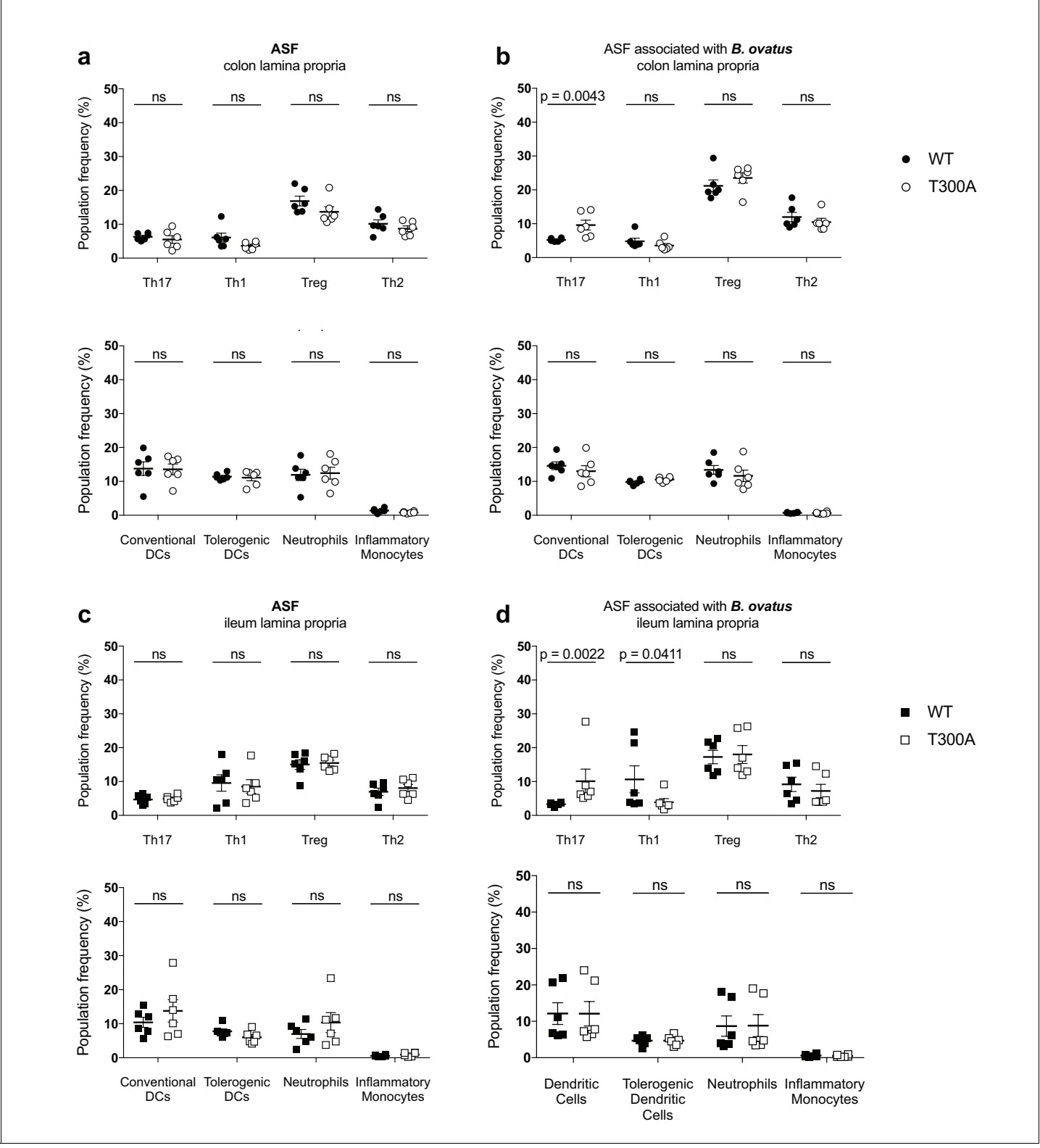

**Figure 8.** Increased frequency of Th17 cells in the gut of T300A ASF mice associated with *B. ovatus* 8483. LP flow cytometry of T cell and myeloid cell populations from ASF mice (WT *n* = 6 and T300A *n* = 6) or ASF mice associated with ~10⁹ CFU of *B. ovatus* 8483 for 3 weeks (WT *n* = 6 and T300A *n* = 6). T cells gated as the frequency of RORγt⁺ (Th17), T-bet⁺ (Th1), Foxp3⁺ (Treg) or GATA-3⁺ (Th2) cells out of CD3⁺CD4⁺ cells and myeloid cell gated as the frequency of CD11c⁺MHCII⁺ cells out of CD45⁺ cells (conventional DCs), CD11b⁻CD103⁺ cells out of CD11c⁺MHCII⁺ cells (tolerogentic

*Figure 8 continued on next page*

*Figure 8 continued*

DCs), CD11b⁺GR-1^int cells out of CD45⁺ cells (neutrophils), and CD11b⁺GR-1^hi cells out of CD45⁺ cells (inflammatory monocytes). CD3⁺CD4⁺ cell and myeloid cell populations were gated on single, live, CD45⁺ lymphocytes. (**a**) ASF colonic LP T cells (top) and myeloid cells (bottom) (**b**) ASF +*B. ovatus* 8483 colonic LP T cells (top) and myeloid cells (bottom). (**c**) ASF ileal LP T cells (top) and myeloid cells (bottom) (**d**) ASF +*B. ovatus* 8483 ileal LP T cells (top) and myeloid cells (bottom). Data are plotted as the mean ± SEM. ns = not significant. Mann-Whitney *U* test.

DOI: https://doi.org/10.7554/eLife.39982.018

The following source data and figure supplements are available for figure 8:

**Source data 1.** Source data for *Figure 8*.
DOI: https://doi.org/10.7554/eLife.39982.021
**Figure supplement 1.** Immune infiltration in ASF mice associated with *B. ovatus* occurs in the absence of disease.
DOI: https://doi.org/10.7554/eLife.39982.019
**Figure supplement 2.** No differences in colonization levels of *B. ovatus* 8483 in GF or ASF mice.
DOI: https://doi.org/10.7554/eLife.39982.020

light on the etiology of CD and provide insight into the relationship between SNPs, dysbiosis and the immune response in the gut.

# Materials and methods

## Key resources table

| Reagent type (species) or resource | Designation | Source or reference | Identifiers | Additional information |
|---|---|---|---|---|
| Genetic reagent (*M. musculus*) | T300A | PMID: 24821797 | | Dr. Ramnik Xavier (Broad/MIT/Harvard) |
| Chemical compound, drug | Dextran Sulfate Sodium | MP Biomedicals | 0216011050–50 g | |
| , strain back ground | *Bacteroides ovatus* | ATCC | 8483 | |
| Software, algorithm | Picard suite | https://broadinstitute.github.io/picard/command-line-overview.html | RRID: SCR_006525 | |
| Software, algorithm | MetaPhlAn | PMID 26418763 | RRID: SCR_004915 | |
| Software, algorithm | HUMAnN2 | http://huttenhower.sph. harvard.edu/humann2 | RRID: SCR_016280 | |
| Software, algorithm | Rtsne | https://cran.r-project.org/web/packages/Rtsne/index.html | RRID: SCR_016342 | |
| Software, algorithm | PICRUSt | PMID 23975157 | RRID: SCR_016856 | |
| Software, algorithm | HUMAnN | PMID 22719234 | RRID: SCR_014620 | |
| Software, algorithm | Prism 7.0b | https://www.graphpad.com/scientific-software/prism/ | RRID: SCR_002798 | |
| Software, algorithm | limma | PMID 25605792 | RRID: SCR_010943 | |
| Software, algorithm | QIIME | PMID: 20383131 | RRID: SCR_008249 | |

*Continued on next page*

*Continued*

| Reagent type (species) or resource | Designation | Source or reference | Identifiers | Additional information |
|---|---|---|---|---|
| Antibody | Rat anti mouse CD16/32 (clone 93) | Biolegend | Cat. #: 101310, RRID:AB_2103871 | Flow cytometry (FC), FC receptor block 1:100 |
| Antibody | Rat anti mouse CD45 (clone 30-F11) | Biolegend | Cat. #: 103114, RRID:AB_312979 | FC, PE/Cy7 1:100 |
| Antibody | Armenian hamster anti mouse CD3e (clone 145–2 C11) | Biolegend | Cat. #: 100334, RRID:AB_2028475 | FC, Pacific Blue 1:100 |
| Antibody | Rat anti mouse CD4 (clone GK1.5) | Biolegend | Cat. #: 100414, RRID:AB_312981 | FC, APC/Cy7 1:100 |
| Antibody | Mouse anti mouse GATA-3 (clone L50-823) | BD Biosciences | Cat. #: 560077, RRID:AB_1645303 | FC, Alexa Fluor 488 1:33 |
| Antibody | Mouse anti mouse Foxp3 (clone 150D) | Biolegend | Cat. #: 320007, RRID:AB_492981 | FC, PE 1:40 |
| Antibody | Rat anti mouse RORgt (clone B2D) | Invitrogen | Cat. #: 17-6981-82, RRID:AB_2573254 | FC, APC 1:40 |
| Antibody | Mouse anti mouse T-bet (clone 4B10) | Biolegend | Cat. #: 644806, RRID:AB_1595488 | FC, PerCP/Cy5 1:33 |
| Antibody | Rat anti mouse CD45 (clone 30-F11) | Biolegend | Cat. #: 103126, RRID:AB_493535 | FC, Pacific Blue 1:100 |
| Antibody | Armenian hamster anti CD11c (clone N418) | Biolegend | Cat. #: 117306, RRID:AB_313775 | FC, FITC 1:200 |
| Antibody | Rat anti mouse CD11b (clone M1/70) | Biolegend | Cat. #: 101228, RRID:AB_893232 | FC, PerCP/Cy5 1:200 |
| Antibody | Rat anti mouse I-A/I-E (MHCII) (clone M5/114.15.2) | Biolegend | Cat. #: 107630, RRID:AB_2069376 | FC, PE/Cy7 1:200 |
| Antibody | Rat anti mouse Ly-6G/Ly-6C (GR-1) (clone RB6-8C5) | Biolegend | Cat. #: 108412, RRID:AB_313377 | FC, APC 1:200 |
| Antibody | Armenian hamster anti mouse CD103 (clone 2E7) | Invitrogen | Cat. #: 12-1031-82, RRID:AB_465799 | FC, PE 1:100 |
| Commercial Assay or Kit | Live/Dead Fixable Yellow Dead Cell Stain | Thermo Fisher Scientific | Cat. #: L34959 | FC |
| Commercial Assay or Kit | Quant-iT PicoGreen dsDNA Assay Kit | Thermo Fisher Scientific | Cat. #: P11496 | DNA quantification |
| Commercial Assay or Kit | Nextera XT DNA Library Preparation Kit | Illumina | Cat. #: FC-131–1096 | Sequencing libraries |

*Continued on next page*

*Continued*

| Reagent type (species) or resource | Designation | Source or reference | Identifiers | Additional information |
|---|---|---|---|---|
| Commercial Assay or Kit | Agilent DNA 1000 Kit | Agilent Technologies | Cat. #: 5067–1504 | Insert sizes |
| Other | RNAlater | Sigma | Cat. #: R0901-500ML | Solution for stool sample collection |
| Commercial Assay or Kit | QIAamp 96 PowerFecal QIAcube HT Kit (5) | QIAGEN | Cat. #: 51531 | Stool DNA extraction |
| Commercial Assay or Kit | Allprep DNA/ RNA 96 Kit | QIAGEN | Cat. #: 80311 | Stool DNA extraction |
| Other | IRS solution | QIAGEN | Cat. #: 26000-50-2 | Solution for stool DNA extraction |
| Commercial Assay or Kit | PowerBead Plates, Glass | QIAGEN | Cat. #: 27500–4-EP-BP | Stool DNA extraction |
| Other | Brucella agar sheeps blood/hemin/ vitamin K plates | Hardy Diagnostics | Cat. #: A30 | Bacterial culture plates |
| Commercial Assay or Kit | Rneasy Mini Kit | QIAGEN | Cat. #: 74106 | RNA isolation |
| Commercial Assay or Kit | iScript cDNA synthesis kit | Bio-Rad | Cat. #: 1708891 | cDNA synthesis |

## Mice

All mouse strains employed are on the C57BL/6J background and were sacrificed between 8–13 weeks of age. Generation of the ATG16L1 T300A knock-in mice has been previously described by *Haberman et al. (2014)*. Conventionally-housed mice were kept at the Massachusetts General

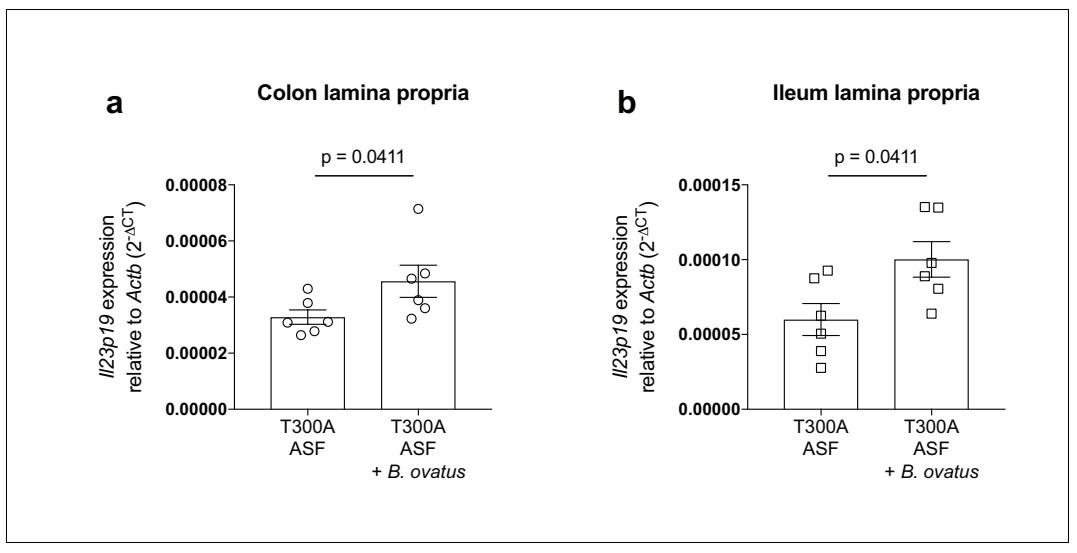

**Figure 9.** Increased expression of *Il23p19* in the lamina propria from the colon and ileum of T300A ASF mice associated with *B. ovatus* 8483. Expression of *Il23p19* by qPCR from colon (**a**) and ileum (**b**) LP cells isolated from ASF T300A mice or ASF T300A mice associated with *B. ovatus* 8483.

DOI: https://doi.org/10.7554/eLife.39982.022

The following source data is available for figure 9:

**Source data 1.** Source data for *Figure 9*.

DOI: https://doi.org/10.7554/eLife.39982.023

Hospital, and all procedures were performed in accordance with the Institutional Animal Care and Use Committee at Massachusetts General Hospital. ATG16L1 T300A knock-in mice were aseptically rederived as gnotobiotic animals and maintained in semi-rigid gnotobiotic isolators under a strict 12 hr light cycle in the Harvard T. H. Chan Gnotobiotic Center for Mechanistic Microbiome Studies. All gnotobiotic experiments were performed at the Harvard T. H. Chan Gnotobiotic Center for Mechanistic Microbiome Studies and were approved and carried out in accordance with Harvard Medical School's Standing Committee on Animals and the National Institutes of Health guidelines for animal use and care. Mice gavaged with different stool samples from human donors were maintained in separate isolator cages. For all mouse experiments a minimum of two experiments were conducted throughout.

## Dextran Sulfate Sodium treatment
Conventionally-housed mice were fed 2.5% (w/v) dextran sulfate sodium (DSS, MP Biomedicals; MW = 36,000–50,000) dissolved in sterile distilled drinking water ad libitum for 7 days, followed by 7 days of regular drinking water. Body weights were monitored daily. Stool samples were collected into RNAlater after the last day of regular drinking water and stored at −80°C until DNA was extracted for 16S rRNA gene amplicon sequencing.

## Mouse stool sample collection and DNA extraction
Stool samples for 16S rRNA amplicon surveys were collected into RNAlater and stored at −80°C prior to DNA extraction. The samples were then stored at −80°C until shipping to the Broad Institute for DNA extraction. For DNA extraction, a combination of the QIAamp 96 PowerFecal Qiacube HT Kit (Qiagen Cat No./ID: 51531), the Allprep DNA/RNA 96 Kit (Qiagen Cat No./ID: 80311), and IRS solution (Qiagen Cat No./ID: 26000-50-2) were used with a custom protocol. One mouse stool pellet per sample was transferred into individual wells of the PowerBead plate, with 0.1 mm glass beads (Cat No./ID: 27500–4-EP-BP) on dry ice. 650 µl of 56°C heated PW1 buffer with 1:100 vol addition of 1M DTT was added directly to each sample well before lysis by bead beating on a TissueLyzer II at 20 Hz for a total of 10 min. Samples pelleted by centrifugation for 6 min at 4500 x g. Supernatants transferred to a new S block (supplied in PowerFecal Kit) and combined with 150 µl of IRS solution and vortexed briefly before 1 min incubation. Sealed samples centrifuged again for 6 min, 4500 x g and up to 450 µl of supernatant were transferred to new S block, combined with 600 µl of Buffer C4 (PowerFecal Kit), mixed by pipetting 10x and incubated at room temperature for 1 min. Samples were transferred into AllPrep 96 DNA plate on top of clean S blocks and centrifuged for 3 min at 4500 x g. Step was repeated until all sample passed through. Allprep DNA plate was placed on top of a 2 mL waste block. 500 µl AW1 buffer was added to the DNA plate which was sealed and centrifuged for 4 min at 4500 x g. 500 µl AW2 buffer was added to the DNA plate, repeating centrifuge step. Allprep DNA plate was placed on top of elution plate. 100 µl of 70°C heated EB Buffer was added to each sample column and incubated for 5 min and centrifuged for 4 min at 4500 x g to elute. Purified DNA was stored at −20°C.

## Human stool sample collection for gnotobiotic mouse association studies
Human stool samples for association with gnotobiotic mice were acquired from patients from Massachusetts General Hospital according to institutional review board #2004P001067. Patients produced the sample at the hospital visit, and it was immediately stored at −80°C by the staff. Participants provided informed consent for the study and are part of the PRISM cohort study.

## Gnotobiotic mouse association with human stool
Frozen human stool samples (50 mg/mouse) were reconstituted in sterile PBS 0.05% Cysteine (250 mg/ml). 25 mg (100 ul) was orally instilled followed by 12.5 mg (50 ul) placed on the anus and 12.5 mg (50 ul) placed on the back fur of each mouse. Stool samples and gut tissues were collected at 4 weeks post human stool association.

## Gnotobiotic and ASF mouse association with bacteroides ovatus 8483

*Bacteroides ovatus* 8483 (ATCC) was grown in liquid culture with *Bacteroides* basal medium as described in Pantosti et al (*Pantosti et al., 1991*). GF or ASF mice were gavaged with ~$10^9$ CFU in 100 µl. Mice were sacrificed after 3 weeks. At sacrifice, stool samples were collected and weighed and CFU/g of stool was calculated after growth on Brucella agar plates with 5% sheep blood, hemin, and vitamin K1 (Hardy Diagnostics A30).

## Sequencing and analysis of the 16S Gene

16S rRNA gene libraries targeting the V4 region of the 16S rRNA gene were prepared by first normalizing template concentrations and determining optimal cycle number by way of qPCR. Two 25 µl reactions for each sample were amplified with 0.5 units of Phusion with 1X High Fidelity buffer, 200 µM of each dNTP, 0.3 µM of 515F (5'-AATGATACGGCGACCACCGAGATCTACACTATGGTAATTG TGTGCCAGCMGCCGCGGTAA-3') and 806rcbc0 (5'CAAGCAGAAGACGGCATACGAGATTCCC TTGTCTCCAGTCAGTCAGCCGGACTACHVGGGTWTCTAAT-3'). 0.25 µl 100x SYBR were added to each reaction and samples were quantified using the formula 1.75(deltaCt). To ensure minimal over-amplification, each sample was diluted to the lowest concentration sample, amplifying with this sample optimal cycle number for the library construction PCR. Four 25 µl reactions were prepared per sample with master mix conditions listed above, without SYBR. Each sample was given a unique reverse barcode primer from the Golay primer set (*Bloom et al., 2011* and *Koropatkin et al., 2012*). Replicates were then pooled and cleaned via Agencourt AMPure XP-PCR purification system. Purified libraries were diluted 1:100 and quantified again via qPCR (Two 25 µl reactions, 2x iQ SYBR SUPERMix (Bio-Rad, REF: 1708880 with Read 1 (5'-TATGGTAATT GT GTGYCAGCMGCCGCGG TAA-3'), Read 2 (5'-AGTCAGTCAG CC GGACTACNVGGGTWTCTAAT-3'). Undiluted samples were normalized by way of pooling using the formula mentioned above. Pools were quantified by Qubit (Life Technologies, Inc.) and normalized into a final pool by Qubit concentration and number of samples. Final pools were sequenced on an Illumina MiSeq 300 using custom index 5'-ATTAGA WACCCBDGTAGTCC GG CTGACTGACT-3' and custom Read one and Read two mentioned above. 16S data were processed using QIIME (RRID:SCR_008249), and taxonomy was assigned using the Greengenes predefined taxonomy map of reference sequence OTUs to taxonomy (*McDonald et al., 2012*). The resulting OTU tables were checked for mislabeling[71] and contamination (*Knights et al., 2011b*). A median sequence depth of 24,583 per sample was obtained, and samples with fewer than 5000 filtered sequences were excluded from analysis.

## Library construction

16S rRNA gene libraries were constructed as previously described (*Kostic et al., 2015*)

## Metagenomics

DNA samples were quantified by Quant-iT PicoGreen dsDNA Assay (Life Technologies) and normalized to a concentration of 50 pg/µl. Illumina sequencing libraries were prepared from 100 to 250 pg of DNA using the Nextera XT DNA Library Preparation kit (Illumina) according to the manufacturer's recommended protocol, with reaction volumes scaled accordingly. Prior to sequencing, libraries were pooled by collecting equal volumes (200 nl) of each library from batches of 96 samples. Insert sizes and concentrations for each pooled library were determined using an Agilent Bioanalyzer DNA 1000 kit (Agilent Technologies). Libraries were sequenced on HiSeq 2 × 101 to yield ~10 million PE reads. Post-sequencing de-multiplexing and generation of BAM and Fastq files are generated using the Picard suite (RRID:SCR_006525) (https://broadinstitute.github.io/picard/command-line-overview. html). Metagenomic data were analyzed using MetaPhlAn (RRID:SCR_004915) (v.2.2) (*Truong et al., 2015*) for taxonomic profiling and HUMAnN2 (RRID:SCR_016280 (http://huttenhower.sph. harvard. edu/humann2) for functional profiling. All sequencing data generated for these studies has been deposited at SRA #SUB4222585

## Principal coordinate and bacterial heatmap plots

Principal coordinate plots were generated using t-Distributed Stochastic Neighbor Embedding (t-SNE) as implemented in the R package Rtsne (RRID:SCR_016342). Bray-Curtis dissimilarity, where xsi denotes the abundance of strain s in sample i, was used as the distance measure. We followed the

guidelines given by the authors (FAQ at http://lvdmaaten.github.io/tsne/) and selected the free parameter, perplexity, by generating mappings with perplexity values between 5 and 50 in increments of 5. Mappings with lowest error and best visual properties were obtained using perplexity = 50. We set the absent proportion data to zeros and remapped the proportions (0, 1) to the interval (1e-7, 1 - 1e-7). We transformed the proportions using logit transformation, after which the data fit a normal distribution.

## PICRUSt analysis

Microbial functional modules were inferred from the 16S rRNA gene amplicon sequencing using PIC-RUSt (*Langille et al., 2013*) (Phylogenetic Investigation of Communities by Reconstruction of Unobserved States) and functional modules were reconstructed using HUMAnN (*Abubucker et al., 2012*) (RRID:SCR_014620).

## Lamina Propria immune cell isolation

Four weeks post association with human stool, the colon and distal 10 centimeters of the small intestine (SI) were removed, opened longitudinally, and rinsed in PBS (Dulbecco's – calcium and magnesium free) to remove fecal contents. The epithelial layer was removed using two rounds of 5 mM EDTA (10 ml/colon 25 ml/SI) rotating at 37C (round 1 includes 1 mM DTT). After removing the epithelial layer, the lamina propria was washed in PBS and chopped into ~1 mM pieces. The tissue was digested in RPMI with glutamine (Sigma) with the following: 10% FBS, 1% penicillin/streptomycin (Corning), 0.5 mg/ml Dispase (STEM cell tech), 1 mg/ml collagenase D (Roche), 50 μg/ml DNAse (Roche) in two rounds of 30 min rotating at 37C. Isolated single cells were filtered over 40 μM filter and resuspended in 5 mM EDTA followed by a wash with FACS buffer (2% FBS 1 mM EDTA). Cells were counted on the hemocytometer followed by flow cytometry analysis.

## Flow Cytometry

$1 \times 10^6$ isolated colon lamina propria or small intestine lamina propria cells were stained with LIVE/DEAD$^{TM}$ fixable yellow dead cell stain kit (Thermo Fisher Scientific, L34959) for 15 min at room temperature. Cells were washed with FACS buffer and then stained with FC block (RRID:AB_2103871, Biolegend, 101310) for 10 min on ice followed by mouse extracellular fluorochrome-conjugated antibodies against mouse: CD45 (RRID:AB_312979 and RRID:AB_493535), CD3 (RRID:AB_2028475), CD4 (RRID:AB_312981), CD11c (RRID:AB_313775), CD11b (RRID:AB_893232), MHCII (RRID:AB_2069376), GR-1 (RRID:AB_313377), and CD103 (RRID:AB_465799). Cells were permeabilized and fixed using the Foxp3 Fix/Perm kit (Biolegend, 421403) and stained at room temperature for 45 min with the following intracellular antibodies against mouse: RORγt (RRID:AB_2573254), T-bet (RRID:AB_1595488), Foxp3 (RRID:AB_492981), or GATA-3 (RRID:AB_1645303). Flow cytometry was conducted using a BD LSR II.

## Colitis score

Gnotobiotically-housed mice associated with stool from human donors were sacrificed and colons were removed, opened longitudinally, placed in cassettes and fixed in 4% paraformaldehyde (Sigma-Aldrich) and embedded in paraffin. Sections (5 μM) were H and E-stained and evaluated in a blinded fashion for epithelial hyperplasia (0–3), epithelial injury (0–3), polymorphonuclear infiltration (0–3) and mononuclear infiltration (0–3).

## Statistical analysis

Statistical analysis for flow cytometry data was performed in Prism 7.0b (RRID:SCR_002798). Data are plotted as mean ± SEM and either Mann-Whitney *U* test or two-way ANOVA as described in figure legend. $p < 0.05$ was considered significant. ns = not significant. To discover association of bacterial abundance and the mouse genotype, we use limma (RRID:SCR_010943), an R package for linear modeling that powers differential expression analyses (*Ritchie et al., 2015*). We adjusted for age, cage, and sex in the analysis. Corrected P-values were generated using Benjamani-Hochberg false discovery rate.

## qPCR

Expression of *Il23p19* was analyzed from total cells isolated from the colon and ileum lamina propria. Epithelial cells were removed as described in the lamina propria immune cell isolation methods described above. Lamina propria cells were resuspended in RLT buffer and frozen at −80C until further analysis. RNA was extracted using the RNeasy Mini kit (QIAGEN) and cDNA was generated using iScript cDNA synthesis Kit (Bio-Rad, 1708891). Quantitative PCR was conducted using the following primers for *Il23p19*: forward primer: AGCGGGACATATGAATCTACTAAGAGA, reverse primer: GTCCTAGTAGGGAGGTGTGAAGTTG.

# Acknowledgements

The authors thank the members of the Garrett lab for their thoughtful discussion, and Tiffany Poon of the Broad Institute and Elizabeth Andrews of MGH for coordinating the samples and patient cohorts. We would also like to thank the Broad Institute Microbial 'omics core, Genomics Platform and Broad Technology Labs. We would like to thank Elizabeth A Creasey for animal husbandry and sample collection. We would also like to thank the Harvard T H Chan Gnotobiotic Center for Mechanistic Microbiome Studies. These studies were supported by RO1 CA154426 (WSG), DK097485 (RJX), DK092405 and R24 DK110499 (RX and WSG), DK105653 (SL) and a Smith Family Foundation Award for Excellence in Biomedical Research to ADK

# Additional information

## Competing interests

Wendy S Garrett: Senior editor, *eLife*. The other authors declare that no competing interests exist.

## Funding

| Funder | Grant reference number | Author |
|--------|------------------------|--------|
| National Institutes of Health | CA154426 | Wendy S Garrett |
| National Institutes of Health | DK092405 | Wendy S Garrett<br>Ramnik J Xavier |
| National Institutes of Health | DK105653 | Sydney Lavoie |
| Richard and Susan Smith Family Foundation | Smith Family Foundation Award for Excellence in Biomedical Research | Aleksander Kostic |
| National Institutes of Health | DK097485 | Ramnik J Xavier |
| National Institutes of Health | DK110499 | Wendy S Garrett<br>Ramnik J Xavier |

The funders had no role in study design, data collection and interpretation, or the decision to submit the work for publication.

## Author contributions

Sydney Lavoie, Conceptualization, Data curation, Formal analysis, Validation, Investigation, Visualization, Methodology, Writing—original draft, Writing—review and editing; Kara L Conway, Conceptualization, Data curation, Investigation, Methodology, Writing—review and editing; Kara G Lassen, Conceptualization, Data curation, Formal analysis, Investigation, Methodology, Writing—review and editing; Humberto B Jijon, Jessica K Lang, Data curation; Hui Pan, Data curation, Formal analysis, Investigation, Methodology; Eunyoung Chun, Monia Michaud, Data curation, Investigation; Carey Ann Gallini Comeau, Ashwin Ananthakrishnan, Investigation; Jonathan M Dreyfuss, Formal analysis, Investigation; Jonathan N Glickman, Data curation, Formal analysis, Investigation; Hera Vlamakis, Data curation, Formal analysis, Supervision, Investigation, Visualization, Methodology, Writing—review and editing; Aleksander Kostic, Conceptualization, Data curation, Formal analysis, Funding acquisition, Validation, Investigation, Visualization, Methodology, Project administration, Writing—

review and editing; Wendy S Garrett, Conceptualization, Resources, Data curation, Formal analysis, Supervision, Funding acquisition, Investigation, Visualization, Methodology, Writing—original draft, Project administration, Writing—review and editing; Ramnik J Xavier, Conceptualization, Supervision, Funding acquisition, Investigation, Visualization, Methodology, Project administration, Writing—review and editing

### Author ORCIDs
Sydney Lavoie (iD) http://orcid.org/0000-0003-2890-3634
Kara G Lassen (iD) http://orcid.org/0000-0001-7023-461X
Wendy S Garrett (iD) http://orcid.org/0000-0002-5092-0150

### Ethics
Animal experimentation: Conventionally-housed mice were kept at the Massachusetts General Hospital, and all procedures were performed in accordance with the Institutional Animal Care and Use Committee at Massachusetts General Hospital. ATG16L1 T300A knock-in mice were aseptically rederived as gnotobiotic animals and maintained in semi-rigid gnotobiotic isolators under a strict 12-hour light cycle in the Harvard T. H. Chan Gnotobiotic Center for Mechanistic Microbiome Studies. All experiments were approved and carried out in accordance with Harvard Medical School's Standing Committee on Animals and the National Institutes of Health guidelines for animal use and care.

### Decision letter and Author response
Decision letter https://doi.org/10.7554/eLife.39982.033
Author response https://doi.org/10.7554/eLife.39982.034

## Additional files
### Supplementary files
• Supplementary file 1. Mouse caging and metadata for *Figures 1–8*. Mouse genotype, age (wks), sex, cage, experiment description and figure are listed above. Mice housed together have the same cage number.
DOI: https://doi.org/10.7554/eLife.39982.024

• Transparent reporting form
DOI: https://doi.org/10.7554/eLife.39982.025

### Data availability
Human microbiome sequence data are available as noted in Jostins et al. 2012 and requests for access can be made via the NCBI Genotypes and Phenotypes database (additional details here https://dbgap.ncbi.nlm.nih.gov/aa/wga.cgi?page=login). Mouse microbiome data have been submitted for deposit at NCBI sequence read archive SUB4222585.

The following dataset was generated:

| Author(s) | Year | Dataset title | Dataset URL | Database and Identifier |
|---|---|---|---|---|
| Aleksander Kostic | 2018 | Illumina HiSeq 2000 sequencing of SAMD00080972 | https://www.ncbi.nlm.nih.gov/sra/?term=4222585 | NCBI Sequence Read Archive, 4222585 |

The following previously published datasets were used:

| Author(s) | Year | Dataset title | Dataset URL | Database and Identifier |
|---|---|---|---|---|
| Judy Cho | 2008 | NIDDK IBDGC Crohn's Disease Genome-Wide Association Study | https://www.ncbi.nlm.nih.gov/projects/gap/cgi-bin/study.cgi?study_id=phs000130.v1.p1 | NCBI Genotypes and Phenotypes database, phs000130.v1.p1 |
| Judy Cho | 2012 | NIDDK IBD Genetics Consortium Ulcerative Colitis Genome-Wide | https://www.ncbi.nlm.nih.gov/projects/gap/ | NCBI Genotypes and Phenotypes |

| Association Study | cgi-bin/study.cgi?study_id=phs000345.v1.p1 | database, phs000345.v1.p1 |

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
