## [Decision Letter]

Thank you for sending your article entitled "The Crohn's disease polymorphism, *ATG16L1T300A*, alters the gut microbiota and enhances the local Th1/Th17 response" for peer review at *eLife*. Your article is being evaluated by three peer reviewers, and the evaluation is being overseen by a Reviewing Editor and Tadatsugu Taniguchi as the Senior Editor.

As you will see, there is consensus among the reviewers that the conclusions of the manuscript, if valid, are of great interest to the field and appropriate for publication in *eLife*. The reviewers and the Reviewing Editor all agree that the strength of the manuscript is the approach, where human IBD specimens are transferred into germ-free mice to reveal new gene-microbiome interactions. However, the absence of certain information, conditions, and controls calls into question the conclusions. The reason why certain data is missing in this study is likely due to the technically challenging and time consuming nature of this powerful approach. The reviewers are sympathetic to this challenge, and wish to give the authors an opportunity to respond. The following appear to be the big picture consensus concerns:

1) Analysis of the microbiome composition pre-DSS (or no DSS) is necessary to interpret the findings post-DSS.

2) The rationale for donor selection is unclear and potentially problematic. The non-T300A donors may be key to interpretation of the experiments and establishing baseline. Similarly, the T300A genotype does not appear to have a contribution towards microbiome composition in UC, although we cannot say that for sure with the current analysis. It is unclear mechanistically why it matters whether the sample is from UC vs CD with the same genotype. Also, why is the healthy control heterozygous for T300A? A related issue is the low N number for human donors. These differences between conditions (health status and genotype) may be a reflection of random donor heterogeneity.

3) A more thorough examination of the immune compartment is needed. There is a sense that the authors are finding what they're looking for.

4) Details regarding co-housing, gender, and littermates are missing. Depending on how the experiments were performed, a subset may need to be repeated under appropriate conditions to validate the initial findings.

If certain samples are readily available, and some of the concerns can be addressed through clarification of text rather than time-consuming experiment, then it may be possible to address these issues within the 2-month review period. The danger is that the authors will invest significant resources to improve the manuscript, only to find that the strategy is insufficient. Therefore, the editors and reviewers wish to invite you to respond within the next two weeks with an action plan and detailed timetable for the completion of the additional work. Please clearly state which issues will be addressed through writing versus generating new experimental results. Also, the above 4 points represent a summary of the discussion and we encourage you to carefully read and consider the detailed comments by the reviewers outlined below. We plan to share your responses with the reviewers and then issue a binding recommendation.

*Reviewer #1:*

The Crohn disease risk allele ATG16L1 (*ATG16L1T300A*), which impacts expression of this important protein has known impacts on host immune response and results in decreased autophagy, increased pro-inflammatory cytokine release, and decreased intracellular bacterial clearance. This submitted manuscript addresses the major existing knowledge gaps regarding whether this and other genetic risk loci for IBD can also drive changes in the gut microbiota.

In this interesting work, the authors have studied a gnotobiotic model using patient fecal transfers in mice, bearing this same risk allele *ATG16L1T300A*, and determine the effects on host-microbiota interactions. Well-designed experiments reveal putative increases in certain bacteria – viz. *Bacteroides ovatus* as well as alterations in gut Th1/Th17 cells in *ATG16L1T300A* mice, without colitis possibly prior to disease onset.

Critique:

1) GWAS studies have linked over 230 genetic risk loci to increased IBD risk but this manuscript addresses only one and even when transferring stool from active Crohn disease patient does not cause experimental colitis per se. Further, already published work has shown that NOD2 variants, which may interact in aberrant manner with ATG16L1, predispose to dysbiosis – see Knights et al., 2014.

2) Alterations in mouse fecal microbiota are only noted in in SPF T300A vs. WT mice following on the generation of gut inflammation, as in Figure 1. It appears that the T300A risk allele may specifically enhance *Bacteroides* spp.

3) Data on the basal microbiome are not provided in quiescent "control" state given that T300A mice do not spontaneously develop intestinal inflammation.

4) In Figure 2, concise data indicate stability of microbiome diversity and changes in control and mutant mice, as dictated by human donor stool status. It is unclear why the healthy volunteer is in fact genotype heterozygous T300A (AG), when even the intermediate phenotype appears to impact the microbiome? Could this be addressed?

5) In Figure 3, the authors show increase in *Bacteroides* in T300A mice that received stool from active Crohn disease, and there are little changes noted in mice receiving stool from active UC patient. Test samples are from small numbers (n=1) of patients, as alluded to by authors.

6) Interestingly, T300A selectively appears to impact gut T cell populations in mice receiving Crohn disease stool (Figure 4) albeit there is no disease induction. In all mice, stool from all sources appears to increase T cells, relative to germ free mice but there are no RORγt T cells increments. There is no substantive colitis induced by donor stool transfer in the experiments reported –Figure 6—figure supplement 3.

7) Patient genotype data and stool microbiome data available from Jostins et al. in ref. 7, suggest that T300A in humans conveys a trend to increase *Bacteroides* fragilis, but not robustly with respect to the genus *Bacteroides* and not in a generally significant manner. (Figure 5). The authors suggest that T300A seems to reconfigure the human microbiota and importantly even the presence of a heterozygous genotype displays an intermediate phenotype with respect to *Bacteroides* species.

This current work has certain limitations as noted above, but is an important step as it provides insight into disease subtypes and has identified dysbiosis linked to genetics, which may impact future treatment avenues for IBD patients as well as informing diagnostic tests.

*Reviewer #2:*

The study by Lavoie et al. explores the interaction between host genetics and the microbiome in inflammatory bowel disease using gnotobiotic T300A mice transplanted with genotyped IBD patient fecal material. Conceptually, this is an important area of investigation as previous studies focused primarily on either the microbial communities of IBD patients, or the role of host genes in modulating SPF microbiota in ATG16L1-deficient or T300A mice. The strength with this study is in combining the relevant microbiome samples (T300A CD vs healthy) in a relevant mouse model (T300A knockin mice) – however, several key details are still unresolved:

Major points include:

1) After 7 days of DSS, followed by 7 days of water is there a difference in weight loss or rebound kinetics between WT and T300A mice? Microbiome analysis pre-DSS colitis would be helpful to support the data in Figure 1, and can be provided as supplemental material.

The differences in microbial communities during inflammation between WT vs T300A mice are intriguing. However, during steady-state (no DSS) conditions, how do the microbiome compare? Are the differences observed during DSS colitis also present during naïve conditions? Is the bloom in *Bacteroides* due to DSS and inflammation? The authors state that these "changes occur before the onset of disease, suggesting that ATG16L1 T300A may contribute initially to dysbiosis […]" – if this is the case, do you observe increased *Bacteroides* before DSS colitis induction?

This is important since the following figure (Figure 2) show distinct communities between WT vs T300A recipient mice upon transplantation with CD active microbiota, conceivably in an inflammatory setting analogous to DSS conditions of Figure 1.

2) Along the same lines, do you observe an enrichment of *Bacteroides* spp. in CD active donor samples (in T300A or WT recipients) vs. CD inactive donor samples? The UC controls are informative, but how does CD active compare to CD inactive in respects to *Bacteroides* abundance?

3) The donors from healthy control and CD patients were nicely controlled, with all donors with T300A genotype. Were any experiments performed with T300 donors (healthy/CD active/CD inactive)? If so, how do they compare in the analysis in PCA plots (Figure 2A)? It appears that host genotype (T300A) of donor alone is not sufficient drive changes in the composition of the microbiota, as both WT and T300A mice cluster closely together when colonized with active UC T300A donors but not active CD T300A donors (Figure 2B).

Additionally, the authors note that "the differences in the active CD microbiota in mice are due to the presence of the risk allele T300A as opposed to microbial population drift due to human stool transfer into mice alone" – if this is the case, would the authors predict that T300 donors with active CD result in no differences when transplanted into WT and T300A recipient mice? This control would also be relevant when examining *Bacteroides* abundance to support the claim that "T300A has the potential to drive increases in *Bacteroides* spp.". Perhaps the authors can refer to the data in Figure 5, and look at animals colonized with WT or Het to compare with T300A samples.

4) In Figure 2B, how does the microbiome of WT and T300A recipients (CD or UC) compare to the initial donor sample? Is there more drift in one genotype vs. the other?

5) In Figure 3, is the "enrichment" of *Bacteroides* in T300A mice with active CD microbiota truly due to T300A genotype driving the increase in *Bacteroides*? It's not clear if the abundance is relative or absolute, and can be compared with the Active UC donors, which appears to be at the same level as T300A Active CD mice. If so, are WT mice with active CD better able to control *Bacteroides* abundance?

*Reviewer #3:*

Lavoie et al. investigate the contribution of germline variation in the core autophagy gene Atg16L1 (Thr300Ala) to intestinal dysbiosis in murine models. This approach reveals how variants significantly associated with inflammatory bowel disease can impact the microbiome even in the absence of overt disease pathology. Overall, the study is a thorough description of the changes that occur in the T300A mice, both at the level of flora and T cell phenotypes. While the datasets detailing microbial dysbiosis is novel and compelling, the study remains incomplete in a few aspects (detailed below, by Figure). Addressing these would significantly elevate the impact of the manuscript while furthering our understanding of how the T300A genotype impacts intestinal immune cell make-up. It would also provide the level of insight necessary for publication in *eLife*. The suggestions listed attempt to be mindful of the scope and resources within the current article.

Major comments:

Figure 1:

– Analysis of microbial diversity in animal studies is complex and requires thorough description of the approach used. While the figure and legend provide some level of detail, it would be useful to more clearly state if animals were gender matched, if there was a gender bias observed.

– Why did authors use 3 mice in WT group that were not cage matched to T300A mutants. (Figure 1A)? The Z-scores (and dendograms) may be driven by these 3 controls that are not ideal for a 16S sequencing study. Do the associations remain if only co-housed mice are re-analyzed?

– Given the strong association of the T300A variant with ileal Crohn's disease, why was ileal "fecal" content not assessed in this study? An attempt to perform this analysis, or a clear rationale as to why it wasn't attempted would benefit the reader.

Figure 2:

– Details regarding co-housing and gender(s) used, along with any gender bias if observed, would benefit the reader again. Were all recipients co-housed (either in pairs or other protocols) throughout the study?

Figure 4:

– What are the baseline alterations in gut immune cell populations in conventionally co-housed WT vs. T300A mice (from Figure 1)?

– Is the impact on Th1/Th17 abundance cell autonomous? Would adoptive transfer of T300A T cells into a WT recipient reproduce this phenotype? If not, it is an interesting observation as well since it would implicate additional host factors in driving T cell polarization.

– Authors only assessed specific T cell subsets, while the T300A genotype is germline. How does the T300A genotype impact abundance and/or function of other immune cell subsets relevant to IBD? Presenting data on Treg, NK/NKT, gdTcells or other tissue resident T cells, macrophage/monocyte, dendritic cell and B cell populations would be highly relevant.

– Given the expansion of RORc^+^ CD4^+^ T cells in the ileum lamina propria, are IL-23 producing DCs or myeloid cells overabundant? A simple measurement of IL12/23 family cytokines from intestinal explants in this study may provide the answer.

Figure 5:

– A discussion on how the observed dysbiosis may reflect on ileal flora would be useful here, given that the T300A variant is strongly associated with ileal CD.

---

## [Author Response]

[Editors' note: the authors’ plan for revisions was approved and the authors made a formal revised submission.]

As you will see, there is consensus among the reviewers that the conclusions of the manuscript, if valid, are of great interest to the field and appropriate for publication in eLife. The reviewers and the Reviewing Editor all agree that the strength of the manuscript is the approach, where human IBD specimens are transferred into germ-free mice to reveal new gene-microbiome interactions. However, the absence of certain information, conditions, and controls calls into question the conclusions. The reason why certain data is missing in this study is likely due to the technically challenging and time consuming nature of this powerful approach. The reviewers are sympathetic to this challenge, and wish to give the authors an opportunity to respond. The following appear to be the big picture consensus concerns:1) Analysis of the microbiome composition pre-DSS (or no DSS) is necessary to interpret the findings post-DSS.

We agree with the reviewers’ concerns and think this is an important control missing from the work provided upon initial submission. We have analyzed stool samples by 16S rRNA gene amplicon sequencing from conventionally-housed, specific pathogen free (SPF) WT and T300A knockin mice under steady state conditions without DSS treatment. These data are now found in Figure 1. We can confirm from our data (histology – colitis scores of zero from the colons of SPF T300A mice) that these mice do not present with colitis under these conditions.

2) The rationale for donor selection is unclear and potentially problematic. The non-T300A donors may be key to interpretation of the experiments and establishing baseline. Similarly, the T300A genotype does not appear to have a contribution towards microbiome composition in UC, although we cannot say that for sure with the current analysis. It is unclear mechanistically why it matters whether the sample is from UC vs CD with the same genotype. Also, why is the healthy control heterozygous for T300A? A related issue is the low N number for human donors. These differences between conditions (health status and genotype) may be a reflection of random donor heterogeneity.

To address these concerns, we analyzed sequencing data from T300A mice associated with stool from a patient with inactiveCrohn’s disease and without a mutation in the *ATG16L1* gene which we include in the re-submission. These data are found in Figure 3.

Indeed, the composition of the microbiota can vary from person to person and within an individual with IBD during a flare and in remission.

The availability of donor human stool that was collected at clinic and immediately flash frozen was limited. The ideal combinations of appropriately collected sample and individual phenotype and genotype were not available to us.

We have incorporated text into the manuscript that more thoroughly addresses donor selection, sample collection challenges, and sample heterogeneity.

As to why the healthy control is heterozygous for T300A, the guanine to alanine substitution, which converts a threonine to alanine at position 300 in ATG16L1, is a relatively common mutation in healthy individuals. Hampe et al. have shown that the frequency of the G allele is 0.53 in control individuals. We have explained this more clearly and in more detail in the text of the re-submitted manuscript.

3) A more thorough examination of the immune compartment is needed. There is a sense that the authors are finding what they're looking for.

We agree that more subsets of T cells as well as myeloid cells can be altered under these conditions and in the context of IBD. We have additional data on Foxp3+ regulatory T cells, Gata3^+^ Th2 cells as well as a variety of myeloid cell subsets including CD11c^+^MHCII^+^ dendritic cells, CD103^+^ “tolerogenic” dendritic cells, CD11b^+^GR-1^+^ neutrophils and inflammatory monocytes from our gnotobiotic association studies that are now included in Figures 6, 7 and 8 in the re-submitted manuscript.

4) Details regarding co-housing, gender, and littermates are missing. Depending on how the experiments were performed, a subset may need to be repeated under appropriate conditions to validate the initial findings.

We provide information on caging, gender, and littermates as requested by the reviewers in Supplementary file 1.

We carried out the following experiments to ameliorate these concerns and significantly enhance the mechanism we present in the manuscript.

We now include experiments that involve associating germ-free (GF) mice and mice with a limited microbiota (Altered Schaedler Flora – ASF mice) with an individual strain of *Bacteroides ovatus* (ATCC 8483). We completed experiments that involved associating WT and T300A cage-matched, age-matched GF and ASF mice with or without *Bacteroides ovatus*. These data are found in Figure 7 and Figure 8

Data from these experiments address the question of whether T300A mice housed with WT mice show differences in gut immune populations by flow cytometry in the presence of *B. ovatus*. We also analyzed the gut tissue for signs of damage, hyper-proliferation, inflammation and any abnormalities observed on histology-based review of these tissues.

Mechanistically, these experiments reveal the effect of *B. ovatus* alone on the gut immune response. We are focusing on *B. ovatus*, as this species was enriched in T300A mice associated with active Crohn’s disease stool, but not in mice associated with stool from patients with active ulcerative colitis in Figure 4 of our manuscript. These findings correlated with increases in Th17 cells in the gut of T300A mice associated with active Crohn’s disease stool. As mentioned in the manuscript text, previous studies have shown a potential effect of *Bacteroides spp*. on Th17 populations in the gut. These new studies that are now part of the resubmitted manuscript specifically address whether an isolate from this species alone or in concert with the ASF can alter Th17 frequency in the gut of T300A mice.

Summary of experiments and textual revisions in this re-submission:

Experimental Revisions:

a) Inclusion of data on donor input microbiome sample, see Figure 3A.

b) Inclusion of data from DSS experiment, e.g. weight trajectories from T300A and WT mice, see Figure 2E.

c) Inclusion of pre-DSS microbiome analyses and colitis scores of T300A and WT conventional mice, see Figure 1.

d) Inclusion of data on flow cytometry profiles from gnotobiotic experiments on Th2, Treg, and myeloid cells subsets, see Figure 6.

*e) B. ovatus* gnotobiotic experiments with lamina propria immune subset profiling, see Figure 7 and Figure 8.

Textual Revisions:

a) Improved textual explanation of donor selection for gnotobiotic experiments.

b) Inclusion of metadata from gnotobiotic experiments, e.g. mouse gender and caging.

c) Fixed typos mentioned by reviewer number 2.

Reviewer #1:[…] 1) GWAS studies have linked over 230 genetic risk loci to increased IBD risk but this manuscript addresses only one and even when transferring stool from active Crohn disease patient does not cause experimental colitis per se. Further, already published work has shown that NOD2 variants, which may interact in aberrant manner withATG16L1, predispose to dysbiosis – see Knights et al., 2014.

Due to the complex nature of IBD and the abundance of SNPs associated with disease, we assert that it is important to look at individual SNPs separately to carry out mechanistic studies aimed out unraveling disease causality.

2) Alterations in mouse fecal microbiota are only noted in in SPF T300A vs. WT mice following on the generation of gut inflammation, as in Figure 1. It appears that the T300A risk allele may specifically enhance Bacteroides spp.3) Data on the basal microbiome are not provided in quiescent "control" state given that T300A mice do not spontaneously develop intestinal inflammation.

We have responded to reviewer #1’s comments 2 and 3 in the main response number 1 above.

4) In Figure 2, concise data indicate stability of microbiome diversity and changes in control and mutant mice, as dictated by human donor stool status. It is unclear why the healthy volunteer is in fact genotype heterozygous T300A (AG), when even the intermediate phenotype appears to impact the microbiome? Could this be addressed?5) In Figure 3, the authors show increase in Bacteroides in T300A mice that received stool from active Crohn disease, and there are little changes noted in mice receiving stool from active UC patient. Test samples are from small numbers (n=1) of patients, as alluded to by authors.6) Interestingly, T300A selectively appears to impact gut T cell populations in mice receiving Crohn disease stool (Figure 4) albeit there is no disease induction. In all mice, stool from all sources appears to increase T cells, relative to germ free mice but there are no RORγt T cells increments. There is no substantive colitis induced by donor stool transfer in the experiments reported –Figure 6—figure supplement 3.7) Patient genotype data and stool microbiome data available from Jostins et al. in ref. 7, suggest that T300A in humans conveys a trend to increase Bacteroides fragilis, but not robustly with respect to the genus Bacteroides and not in a generally significant manner. (Figure 5). The authors suggest that T300A seems to reconfigure the human microbiota and importantly even the presence of a heterozygous genotype displays an intermediate phenotype with respect to Bacteroides species.

These comments have been addressed in main response number 2 above and in the text of the manuscript as well.

Reviewer #2:[…] 1) After 7 days of DSS, followed by 7 days of water is there a difference in weight loss or rebound kinetics between WT and T300A mice? Microbiome analysis pre-DSS colitis would be helpful to support the data in Figure 1, and can be provided as supplemental material.The differences in microbial communities during inflammation between WT vs T300A mice are intriguing. However, during steady-state (no DSS) conditions, how do the microbiome compare? Are the differences observed during DSS colitis also present during naïve conditions? Is the bloom in Bacteroides due to DSS and inflammation? The authors state that these "changes occur before the onset of disease, suggesting that ATG16L1 T300A may contribute initially to dysbiosis […]" – if this is the case, do you observe increased Bacteroides before DSS colitis induction?This is important since the following figure (Figure 2) show distinct communities between WT vs T300A recipient mice upon transplantation with CD active microbiota, conceivably in an inflammatory setting analogous to DSS conditions of Figure 1.

We have addressed reviewer # 2 concerns in comment 1 in the main response number 1 above.

We include the weight trajectory data for DSS treated mice in the resubmission.

2) Along the same lines, do you observe an enrichment of Bacteroides spp. in CD active donor samples (in T300A or WT recipients) vs. CD inactive donor samples? The UC controls are informative, but how does CD active compare to CD inactive in respects to Bacteroides abundance?3) The donors from healthy control and CD patients were nicely controlled, with all donors with T300A genotype. Were any experiments performed with T300 donors (healthy/CD active/CD inactive)? If so, how do they compare in the analysis in PCA plots (Figure 2A)? It appears that host genotype (T300A) of donor alone is not sufficient drive changes in the composition of the microbiota, as both WT and T300A mice cluster closely together when colonized with active UC T300A donors but not active CD T300A donors (Figure 2B).Additionally, the authors note that "the differences in the active CD microbiota in mice are due to the presence of the risk allele T300A as opposed to microbial population drift due to human stool transfer into mice alone" – if this is the case, would the authors predict that T300 donors with active CD result in no differences when transplanted into WT and T300A recipient mice? This control would also be relevant when examining Bacteroides abundance to support the claim that "T300A has the potential to drive increases in Bacteroides spp.". Perhaps the authors can refer to the data in Figure 5, and look at animals colonized with WT or Het to compare with T300A samples.

We have noted that we included information regarding association of T300A mice with stool from a patient with inactive CD and a WT genotype.

4) In Figure 2B, how does the microbiome of WT and T300A recipients (CD or UC) compare to the initial donor sample? Is there more drift in one genotype vs. the other?

We have 16S rRNA gene amplicon data on human stool samples pre-association that are included in the re-submitted manuscript.

Reviewer #3:[…] Figure 1:– Analysis of microbial diversity in animal studies is complex and requires thorough description of the approach used. While the figure and legend provide some level of detail, it would be useful to more clearly state if animals were gender matched, if there was a gender bias observed.

We provide information on gender in our analysis. We find this topic compelling and in these experiments we did not find differences related to gender in the mice.

– Given the strong association of the T300A variant with ileal Crohn's disease, why was ileal "fecal" content not assessed in this study? An attempt to perform this analysis, or a clear rationale as to why it wasn't attempted would benefit the reader.

We think that because human stool samples are collected as passed fecal samples and not ileal samples, we wanted to remain consistent with how these human samples were collected when we conducted our mouse microbiota studies.

Figure 2:– Details regarding co-housing and gender(s) used, along with any gender bias if observed, would benefit the reader again. Were all recipients co-housed (either in pairs or other protocols) throughout the study?

This comment has been addressed in the main response number 4 above.

Figure 4:– What are the baseline alterations in gut immune cell populations in conventionally co-housed WT vs. T300A mice (from Figure 1)?– Is the impact on Th1/Th17 abundance cell autonomous? Would adoptive transfer of T300A T cells into a WT recipient reproduce this phenotype? If not, it is an interesting observation as well since it would implicate additional host factors in driving T cell polarization.

We think that these experiments, while important, cannot be completed in a two month time frame.

– Authors only assessed specific T cell subsets, while the T300A genotype is germline. How does the T300A genotype impact abundance and/or function of other immune cell subsets relevant to IBD? Presenting data on Treg, NK/NKT, gdTcells or other tissue resident T cells, macrophage/monocyte, dendritic cell and B cell populations would be highly relevant.

We have addressed this with additional data mentioned in the main response number 3 above.

– Given the expansion of RORc^+^ CD4^+^ T cells in the ileum lamina propria, are IL-23 producing DCs or myeloid cells overabundant? A simple measurement of IL12/23 family cytokines from intestinal explants in this study may provide the answer.Figure 5:– A discussion on how the observed dysbiosis may reflect on ileal flora would be useful here, given that the T300A variant is strongly associated with ileal CD.

We did not collect data on IL-23 or collect ileal contents. However, we provide data on Il-23 transcript levels from the lamina propria of the colon and ileum in Figure 9 wherein we show data demonstrating increased expression of *Il23p19* in lamina propria from the colon and ileum of T300A ASF mice associated with *Bacteroides ovatus* 8483.